# Real-time determination of earthquake focal mechanism via deep learning

Wenhuan Kuang [1✉], Congcong Yuan [2] & Jie Zhang [3✉]

An immediate report of the source focal mechanism with full automation after a destructive earthquake is crucial for timely characterizing the faulting geometry, evaluating the stress perturbation, and assessing the aftershock patterns. Advanced technologies such as Artificial Intelligence (AI) has been introduced to solve various problems in real-time seismology, but the real-time source focal mechanism is still a challenge. Here we propose a novel deep learning method namely Focal Mechanism Network (FMNet) to address this problem. The FMNet trained with 787,320 synthetic samples successfully estimates the focal mechanisms of four 2019 Ridgecrest earthquakes with magnitude larger than Mw 5.4. The network learns the global waveform characteristics from theoretical data, thereby allowing the extensive applications of the proposed method to regions of potential seismic hazards with or without historical earthquake data. After receiving data, the network takes less than two hundred milliseconds for predicting the source focal mechanism reliably on a single CPU.

[1] Department of Geophysics, Stanford University, Stanford, CA 94305, USA. [2] Department of Earth and Planetary Sciences, Harvard University, Cambridge, MA 02138, USA. [3] Department of Geophysics, University of Science and Technology of China, Hefei, Anhui 230026, People's Republic of China. ✉email: wenhuan@stanford.edu; jzhang25@ustc.edu.cn

Mitigating the damaging level of earthquake hazards has been a long endeavor in seismology[1–3]. When a destructive earthquake occurs, real-time reporting of the earthquake parameters is of crucial importance for immediate destruction assessment and emergency evacuations. Recent efforts have been refined towards applying artificial intelligence (AI) technologies to estimate the source parameters because of its full automation, high efficiency, and human-like capability[4–6], which has been remarkably demonstrated in numerous seismic processing tasks such as earthquake detection[7,8], seismic phase picking[9–11], magnitude estimation[12], and others[13–18]. Besides reporting the three basic parameters of an earthquake (i.e., origin time, location, and magnitude), it is also exceedingly important to derive the source focal mechanism in time to better understand various aspects of the earthquake. For example, we can use source focal mechanisms to characterize faulting geometry and faulting mechanism[19,20]. A group of focal mechanisms can be used to invert the spatial stress field distribution[21–23]. We can also use the focal mechanism of the mainshock to calculate the static Coulomb stress changes[24–26] for examining the earthquake triggering theory of the aftershocks[27–29]. Furthermore, the timely derived source focal mechanism can provide significant additions such as fault orientation and slipping mode to the point-source ground motion prediction model that is currently in practice[30–32], and thus has the potential to help improve the predicted ground shakings for early warning purpose. The immediate determination of the source focal mechanism is therefore of great importance to monitor and assess seismic hazards.

Compared to determining other source parameters of an earthquake (i.e., origin time, location, and magnitude), the estimation of the source focal mechanism usually requires much more human interactions and lacks full automation and efficiency. Approaches for conventionally resolving the focal mechanism mainly have three categories based upon the waveform information used, such as first motions of P waves[33,34], amplitudes of P and/or S waves[35,36], and full waveforms[37,38]. After receiving the seismic data, these conventional methods usually take a few minutes to tens of minutes for retrieving the focal mechanism solution, and hence incapable of realizing the real-time reporting. Several recent studies first apply deep learning to estimate the P wave first-motion polarities[9,39–41], and then apply the first motions to carry out focal mechanism inversion using programs such as HASH[42]. One of such efforts leads to improved focal mechanisms in California compared to existing catalogs[9]. Several seismological studies suggest that utilizing waveform data can provide better constraints for deriving the focal mechanism than using the P wave first-motion polarities[37,38,43]. More importantly, our objective is to develop a seamless real-time solution for obtaining the focal mechanism in a fully automated fashion. Directly obtaining the focal mechanism of an event from waveform data with processing effort as little as possible is more appealing. Another recent progress that took advantage of an advanced search engine was performed[44] to estimate earthquake source parameters in less than 1 s. Although this approach reduces the time cost significantly, its implementation may be infeasible and impractical since it requires a tremendous search database (~hundreds of Gigabits) for each upcoming search. Besides, the search engine approach needs to reorganize the recorded waveforms as one-dimensional (1D) super trace and it is infeasible in implementation. Hence, the challenge remains in the full automation and practical implementation for real-time determination of the source focal mechanism.

In this study, we leverage the powerful advances in deep learning and propose a novel deep convolutional neural network (Focal Mechanism Network, FMNet) for estimating the source focal mechanism rapidly using full waveforms. Unlike common applications in which the training of supervised neural network models demand voluminous real data, the proposed FMNet can be trained with synthetic data at first and then applied to real data directly. FMNet learns the universal characteristics of waveforms concerning the source focal mechanisms from the synthetic training data. This considers the scenarios without enough historical source focal mechanisms for training the neural network model, especially for those regions with limited seismicity but having the potential seismic hazards. For generating the large training dataset, we discretize the three-dimensional (3D) grid space of the study area of interest. We simulate theoretical waveforms with a variety of focal mechanisms at each spatial grid point. We train the FMNet model with the synthetic dataset and then apply it to predict the focal mechanisms of four real earthquakes with magnitudes larger than 5.4 of the Ridgecrest sequence which occurred in July 2019 in southern California. Additionally, we produce a by-product of the encoder, which is a sparse representation of the input waveforms, to analyze the working mechanism and robustness of the FMNet.

## Results

**Study area and data preparation.** The study area is located in the region of Ridgecrest in southern California (Fig. 1), where a damaging earthquake sequence proceeded by an Mw 6.4 fore-shock and followed by an Mw 7.1 mainshock in July 2019. Four moderate-to-large earthquakes (Mw > 5.4) in the sequence are selected for this study. We collect the three-component (3-C) seismograms from 16 seismometers that are deployed by the Southern Californian Seismic Network (SCSN) around the Ridgecrest area. They are utilized as the testing data for examining the validity of the proposed FMNet. Before the applications of the FMNet model, sufficient training data are vital for assuring a well-trained neural network. Here, instead of using the historic data, we simulate hundreds of thousands of synthetic data as training data since there are very limited source focal mechanisms of historical earthquakes available in this area.

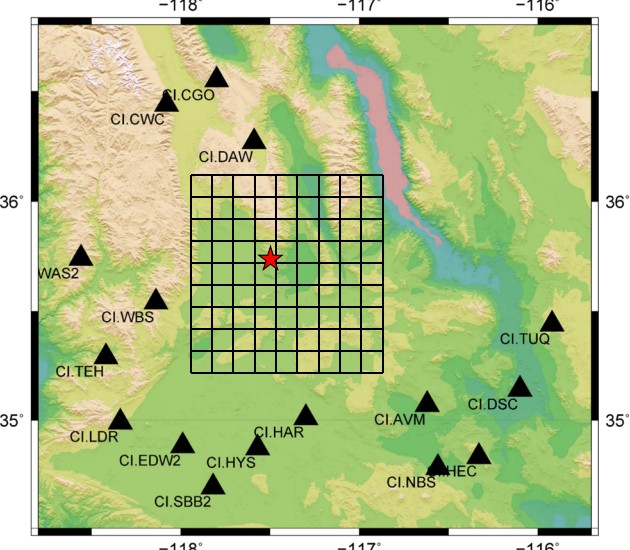

**Fig. 1 Grid discretization of the study area.** The study area is located in the Ridgecrest region of southern California. The range of monitoring area is about 100 km × 100 km in both latitude and longitude directions. The 3D grid discretization has a depth range from 2 to 20 km. In all, 16 seismic stations (black triangles) within 150 km are used to model the 3-C synthetic training data. Red star denotes the mainshock in the Ridgecrest sequence.

As shown in Fig. 1, the study area is discretized from 35.4° to 36.2° in latitude direction, from −118.0° to −117.2° in longitude direction, and from 2 to 20 km in depth. The intervals are 0.1°, 0.1°, and 2 km for latitude, longitude, and depth, respectively. We have $9 \times 9 \times 10 = 810$ virtual grid locations in 3D space. Assuming a double-couple source model[45] and a 1D velocity model of southern California[46], we simulate the 3-C waveforms at 16 seismic stations by adopting the Thompson-Haskell propagator matrix[47]. For each virtual 3D grid, we simulate synthetic waveforms for all combinations of the strike, dip, and rake angles in the ranges of 0° to 360°, 0° to 90°, and −90° to 90°[37], respectively. The used intervals of the strike, dip, and rake angles are 30°, 10°, and 20°, respectively. Hence, we have $12 \times 9 \times 9 = 972$ focal mechanisms for each virtual grid and overall $810 \times 972 = 787,320$ synthetics as training samples, of which each sample contains the 3-C waveforms of 16 seismic stations with a time length of 128 s. Since we normalize the waveforms of each synthetic sample based on the maximum amplitude, we choose a fixed magnitude for all events when modeling the synthetic waveforms. We use 1 s as the sampling rate in all the simulations. Therefore, each training sample has a size of 48 (3 components × 16 stations) × 128 (data length). Additionally, we have prepared another 1000 synthetic samples as the validation dataset. The validation dataset serves as unseen data to evaluate the training performance.

The training samples are processed by filtering between 0.05 and 0.1 Hz, aligning with the theoretical P wave first arrivals, and normalizing to the maximum amplitude. For real data, we need to take the picked onset time of each trace for carrying out the waveform alignment. These preprocessing procedures are important because they help us get rid of the effect from other source parameters such as location and magnitude, and mitigate the dependence on the heterogeneity of velocity media. Considering the real data may present noise and picking errors, we generate realistic scenarios by adding realistic noise and a random time shift (<2 s) to the synthetics (Supplementary Fig. 1) since advanced techniques have greatly reduced the picking errors[9–11]. The realistic noise is extracted from the real recordings at each seismic station. When adding the realistic noise, we randomly scale the amplitudes of noise to account for different signal-to-noise ratios (SNR). The random time shifts are added to each trace of the training samples to account for the picking errors. We process all the synthetic data in the same way and use them to train the network. After the FMNet is well trained, in case that one real earthquake is identified with the existing algorithms of automatic detection and phase picking[7–11], we first remove their instrument responses and then perform the bandpass filtering, arrival-time alignments, and amplitude normalizations on the data prior to feeding them to the FMNet.

**FMNet training and prediction**. The framework of the real-time determination of the source focal mechanism is presented in Fig. 2. It consists of two parts: FMNet training and prediction. For the training part, we train the FMNet with the synthetic data prepared previously along with the corresponding training labels. We describe the architecture of the FMNet, training labeling, and the associated training parameters in the Methods section. In the training process, both the training and validation losses, and the goodness of fitting between true and predicted labels of validation data are viewed as metrics to evaluate the performance of the training process (Supplementary Figs. 2 and 3). The stabilized training and validation loss curves after 50 iterations with sufficiently low resultant values and the high fitting level of between true and predicted labels, both indicate that the FMNet has been stably trained. When it comes to the prediction part, we can directly feed the processed recordings of a real earthquake into the trained FMNet to predict the source focal mechanism. The training process takes about 5 h with 4 GPUs of NVIDIA Tesla V100 for acceleration. However, once well trained, the designed FMNet can output a focal mechanism solution in only 196 ms on a single CPU. Moreover, the trained network model can be deployed to estimate the source focal mechanisms in areas of interest permanently.

To evaluate the general performance and estimate the errors of our model, we generate another test dataset of about 1000 unseen synthetic samples simulated with a diversity of focal mechanisms of normal, strike-slip, and reverse faulting mechanisms (Supplementary Fig. 4). Using the well-trained model, we predict the focal mechanisms on this test dataset. For these predicted focal mechanisms, we adopt the Kagan angle analysis[48,49] to quantify the estimation errors, in which each Kagan angle quantitatively characterizes the difference in rotation angle between the true and the predicted focal mechanisms. From the results of Kagan angle distribution (Supplementary Fig. 5), we find that 97.8% of the Kagan angles are within 20° and only a small fraction of about 2.2% of the estimates have an error larger than 20°. Investigating the remaining 2.2%, we find it is probably caused by the equivalent property of the two nodal planes of the focal mechanism (the true and the auxiliary nodal planes are equivalent). Including more constraints such as first-motion polarities should possibly mitigate this issue and further improve the model. Nevertheless, this test shows that our model can stably predict most events (97.8%) on testing a variety of unseen data with acceptable estimation errors. Furthermore, this test also validates that our model has learned the general ability to predict a diversity of focal mechanisms.

**FMNet prediction results**. The source focal mechanisms of four large earthquakes (Mw > 5.4) in the Ridgecrest sequence are estimated with the trained FMNet. We show these results as red beach balls in Fig. 3. The predicted focal mechanisms generally reveal the strike-slip faulting with very steeply dipping fault planes. Among them, the three focal mechanisms in the southern region, including the Mw 6.4 foreshock and Mw 7.1 mainshock, demonstrate pressure axes in the north–south direction and tension axes in the east–west direction. The other one in the northernmost region shows a slight rotation in the fault plane azimuth. For comparison, we also plot the focal mechanism results from the SCSN moment tensor catalog as reference solutions (in black) in Fig. 3. We can see that the predicted focal mechanisms by the FMNet and the reference focal mechanisms from the SCSN catalog are essentially consistent for the three earthquakes in the southern region, considering the differences in methods, parameterization, velocity model, and the number of recording stations used. The northernmost event is not included in the SCSN catalog for comparison. For this event, we conduct the widely used generalized cut-and-paste method (gCAP)[38] to invert its focal mechanism as shown in gray. We observe that the inverted focal mechanism and the predicted focal mechanism match well for this event. Moreover, the slight rotation of fault azimuth is consistent with the distribution pattern of the aftershock event locations (gray dots). Comparing to other studies regarding this earthquake sequence[20,23,50], the predicted focal mechanisms by our FMNet are essentially consistent with previous results. All these results demonstrate that the proposed FMNet enables us to determine the source focal mechanisms effectively. Additionally, the trained FMNet only takes 196 ms with a minimum requirement of computing resources and memory storage, which outperforms both the conventional methods and the fast search method.

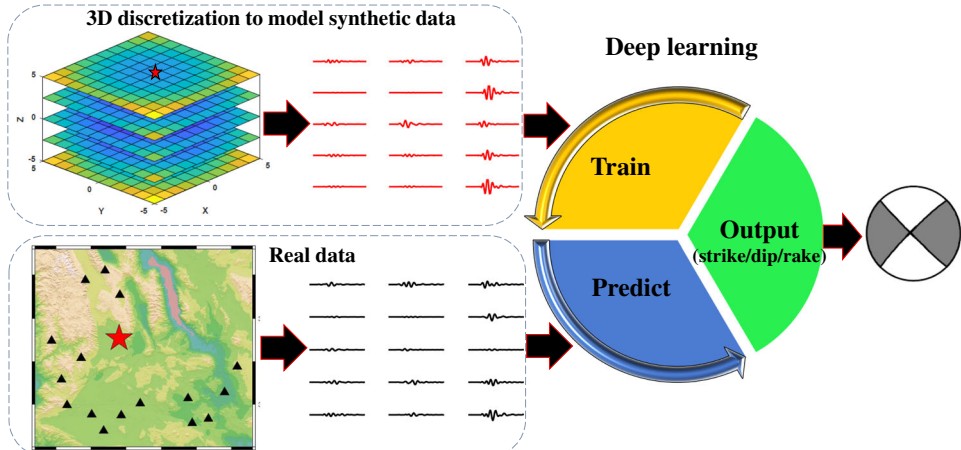

**Fig. 2 Schematic flowchart illustrating the framework of determining source focal mechanism via deep learning.** First, we discretize the monitoring area of interest into three-dimensional (3D) grids and simulate the theoretical waveforms (red waveforms) as training data to train a designed FMNet. Then, when a real earthquake occurs (red star), we directly feed the recorded waveforms (black waveforms) into the well-trained FMNet and output the earthquake focal mechanism directly.

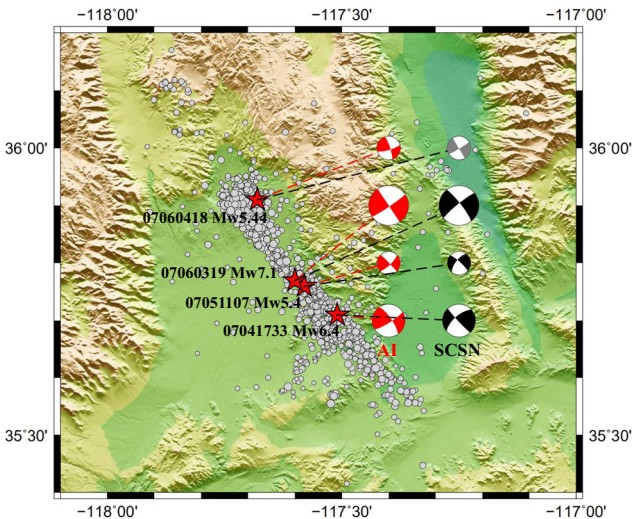

**Fig. 3 The application results to the July 2019 Ridgecrest sequence in southern California.** Four large earthquakes with magnitudes larger than 5.4 (red stars), including the foreshock of Mw 6.4 and the mainshock of Mw 7.1, are tested. The determined focal mechanisms from artificial intelligence (AI) are shown in red, and the reference solutions from Southern California Seismic Network (SCSN) moment tensor catalog are shown in black for comparison. The northernmost focal mechanism (gray) is inverted using gCAP method since it is not included in SCSN moment tensor catalog. Gray dots show background seismicity.

The comparison of waveforms is the most straightforward way to evaluate the predicted results. For this purpose, we simulate the synthetic waveforms using the predicted source focal mechanisms by our FMNet and analyze the similarity between real waveforms and synthetic waveforms (Supplementary Fig. 6). After comparison, we find that both the amplitude and phase information of waveforms across different seismic stations are overlapped well and the computed cross-correlation coefficients reach 0.86, which indicates that the FMNet has learned the ability to recognize the waveforms and mapping them to the corresponding source focal mechanism solution reliably.

**Interpreting the FMNet using the encoder.** To further investigate the working mechanism of our FMNet, by adopting an idea

developed for face recognition, where the network learns a mapping from face images to a compact Euclidean space where distances directly correspond to a measure of face similarity[51,52], we output the extracted features to analyze the reliability and robustness of our model. The last layer of the compression part of the FMNet is exported as a by-product of the encoder (see Fig. 6 and Methods section for details). After training, this encoder can take any training input with the size of $1 \times 48 \times 128$ and output the extracted feature with the size of $128 \times 1 \times 1$. With the encoder, we verify the hypothesis that a measure of feature similarity in feature domain is equivalent to a measure of waveform similarity in data domain through adopting the following steps: First, we calculate the extracted features using the encoder for the whole training dataset to build an encoded database in feature domain. Then, we calculate the extracted features of the data that records a real earthquake. Finally, we measure the L2-norm misfits between the encoded database of training data and the encoded features of the real data in feature domain. For comparison, we also calculate the L2-norm misfits in data domain measuring the waveform differences between real data and training database. By finding the smallest L2-norm misfit, if the retrieved best solution in feature domain corresponds to the best solution retrieved in data domain, we can therefore validate the above hypothesis.

We take the Mw 6.4 foreshock as an example. With the steps illustrated above, we display the comparison of L2-norm misfit distributions that are calculated in data domain (in red) and in feature domain (in black) in Fig. 4a, after ranking in ascending order. Since the whole training dataset is too large, we plot only the first 5000 smallest misfits for clarification. We can see that the L2-norm misfit distributions calculated in data domain and feature domain present a similar shape. Meanwhile, Fig. 4b–d show the corresponding training labels of the strike, dip, and rake angles, respectively, for the L2-norm misfits in feature domain (the black curve in Fig. 4a). By finding the smallest L2-norm misfit, the retrieved best solution of the strike, dip, and rake angles in feature domain are highlighted as magenta circles. Then we compare the best solutions retrieved in data domain and feature domain as shown in Fig. 5. We can observe that the best solution retrieved in feature domain (in magenta) matches well with the best solution that is retrieved in data domain (in red). These analyses and comparison results validate our hypothesis that the extracted features in feature domain maintain the

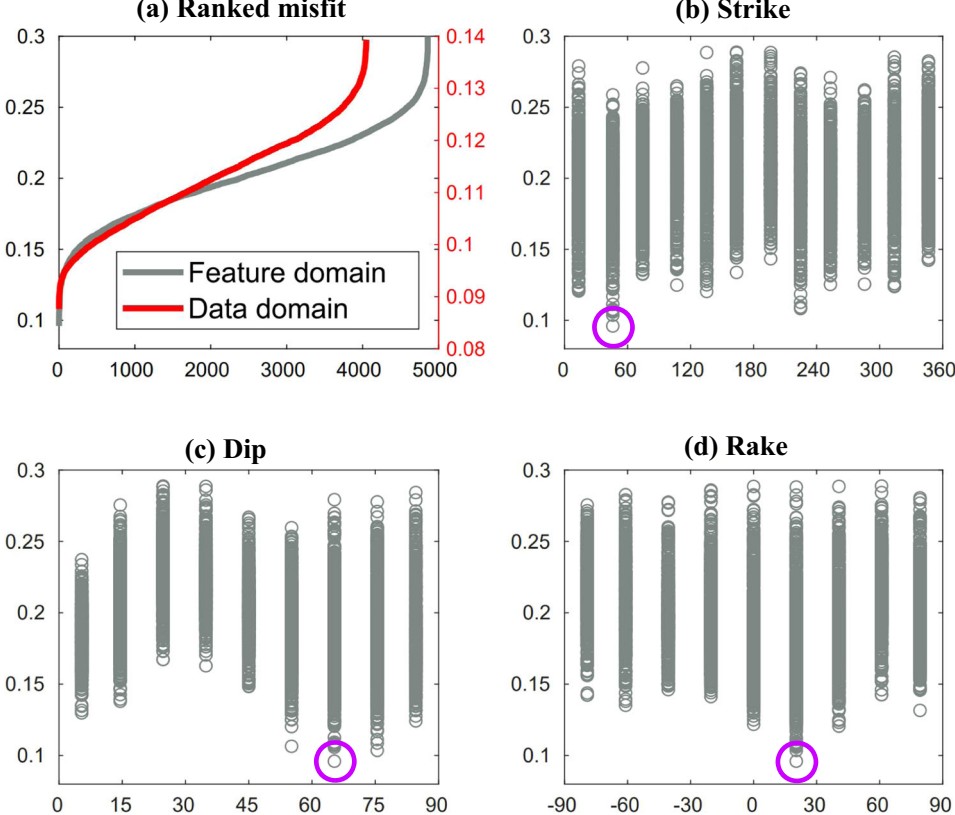

**Fig. 4 Interpreting the neural network using the encoder. a** The comparison of the L2-norm misfit distributions that are calculated in data domain (in red) and feature domain (in black) for the Mw 6.4 foreshock, which are ranked in ascending order. For clarification, only the first 5000 smallest misfits are plotted. **b–d** The corresponding training labels represented in the discretized strike, dip, and rake angles, for the L2-norm misfit distribution in feature domain as shown in the black curve in (**a**). The best solutions of the strike, dip, and rake angles retrieved by finding the smallest L2-norm misfit are highlighted in magenta circles.

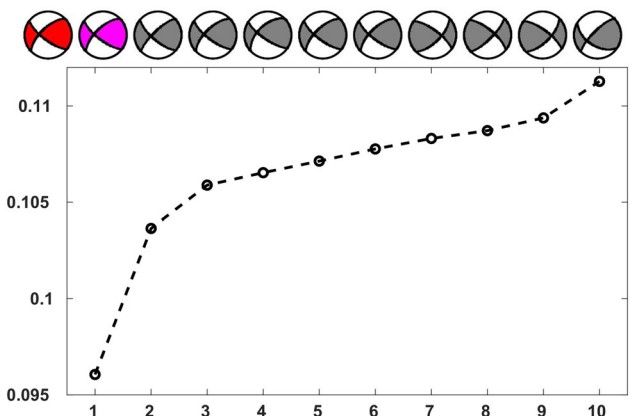

**Fig. 5 The comparison of best solutions retrieved in data domain (in red) and feature domain (in magenta).** The 10 best solutions (in magenta and black) with the smallest L2-norm misfits retrieved in feature domain are also shown for comparison after ranked in ascending order.

essential information of the original waveforms in data domain under the least-square sense, thus the extracted features are sufficient to identify its corresponding source focal mechanism. Moreover, the 10 best solutions (in magenta and black) retrieved in feature domain are generally consistent with minor variations, which illustrates the stability of the trained network.

From the above analysis, the compression part of our FMNet (i.e., the encoder) can be interpreted as a sparse transformation of the input waveforms, where the input data have been compressed

from $1 \times 48 \times 128$ to $128 \times 1 \times 1$ in size by a decreasing factor of 48 times while keeping the key information in the data. The encoder also provides an alternative way to rapidly retrieve the best-matched source focal mechanism by searching in the dataset with encoded features that are prepared with the training data in advance. The expansion part of the FMNet mainly takes these extracted features to reconstruct a mapping function that yields the Gaussian distributions to represent the three angles of a focal mechanism. We address all these analyses presented in this section are for understanding the working mechanism of the FMNet and also for robustness analysis. When the proposed deep learning methodology is applied in a real case, we can directly feed the real data into the well-trained FMNet and output the focal mechanism rapidly. The intermediate output of the extracted feature maps can be used to further evaluate the reliability of the solution.

## Discussion

The proposed FMNet is a deep-learning-based intelligent algorithm that allows us to estimate the source focal mechanism using data from a specific seismic network. The FMNet extracts and learns the essential features of the waveforms from the training dataset, thereby memorizing all the information into the neural network, and hence re-visiting the database is unnecessary. The proposed FMNet only stores the neural weights of a few Mega-bytes for the memory usage, which enables the FMNet to be feasible for automatic real-time applications. In earthquake monitoring, it may take several tens of seconds to over a minute for receiving the full waveform data needed at a number of

seismic stations depending on the source–station distance. From that point, our neural network system takes about an additional 200 ms for analysis, which is negligible.

For resolving the focal mechanism of an earthquake, we are able to train the FMNet with a synthetic waveform dataset. Therefore, we can apply the proposed method to areas with low seismicity but high risks of potential earthquake hazards. However, the calculation of synthetics requires an accurate velocity model corresponding to the frequency band of data. In the case of southern California, this is not a major concern since velocity models have been well studied. For other areas, it may require substantial effort on modeling and inversion to ensure an accurate velocity model before applying the approach. We present a numerical study using a velocity model with perturbations (Supplementary Fig. 7). We perturb the true velocity model by a maximum of 10% in each layer to generate a different model. From the testing results (Supplementary Fig. 8), we can tell that the estimation errors for dip and rake are 8° and 20°, respectively, and their prediction probabilities are lowered as well.

Approach using the P wave first motions requires sufficient azimuthal station coverage to constrain the focal sphere, however, inverting three-component waveform data from a single seismic station should be able to resolve the entire source focal mechanism in theory[53–56]. The FMNet utilizes the three-component full waveform information and it shows smaller estimation errors compared to the approach based on the P wave first motions (Supplementary Figs. 9 and 10). The deep learning approach is different from inversion. When some of the stations do not offer data for input, missing data are replaced by zeros as input while the training data still keep the full set. We design a test with the distribution of the halved number of stations on one side of the event and the rest of the stations with zero traces for data (Supplementary Fig. 11). The event is assumed to occur in an area with training data available. From the test results, we find that the strike and dip angles are well resolved, but the rake angle is off by nearly 20°, and the prediction probability of rake is significantly lower (about 0.5) (Supplementary Fig. 12). Therefore, it is important to evaluate the prediction probabilities.

Since we use synthetics associated with a 1D velocity model to create a dataset for training and testing, it limits the application to low-frequency data, which are generally available from moderate and large events. In this study, the FMNet is evaluated on four earthquakes with magnitudes larger than 5.4. Hundreds of earthquakes below Mw 5.4 in the same area could not be used for testing because of missing low-frequency signals in the data or poor data quality. Further development efforts are needed to

combine the P wave first motions and waveform data to handle smaller events. Generating a 3D velocity model with great details could help model the high-frequency data as well.

To further verify our model on the cases with outliers, we test the scenario that some of the recording stations have data issues and waveforms are missing, but the azimuthal coverage is still good (Supplementary Fig. 13). We find that the predicted probability distributions can match well with the true distribution in terms of their shape and maximum values when partial data are missing (Supplementary Fig. 14). We also test a case where an event occurs out of the study area (Supplementary Fig. 15). The test results show that the predicted probability is much smaller (about 0.6), which can help quantify the reliability of the predicted results (Supplementary Fig. 16). From these test results, we find that an inaccurate velocity model, poor azimuthal coverage, or events out of the network might degrade the prediction performance with low probability. Therefore, using the predicted probability to quantify the reliability of the predicted result is essential.

Seismic waveforms have been often used to model source depth. However, simultaneously returning the focal depth along with parameters of the focal mechanism from the FMNet is unlikely. This is because the sensitivity of the focal depth to waveform data is far smaller than the focal mechanism (Supplementary Fig. 17). After obtaining the focal mechanism, nevertheless, one can find the focal depth by applying a grid search to match waveforms with a fixed focal mechanism.

The current FMNet is designed for monitoring local or regional events within the coverage of a seismic network. Similar to the state-of-the-art methodology for resolving source focal mechanisms by applying moment tensor inversion, the FMNet is limited to moderate and large earthquakes that can be numerically modeled. Developing the capability to simulate waveforms of small earthquakes in high frequency warrants further study. Despite these limitations, the FMNet offers a rapid and reasonably accurate solution to the focal mechanism, a critical component of earthquake information, for which it used to take several minutes in automated earthquake reporting systems, and sometimes it requires quality assurance by humans. With the FMNet, it could potentially help advance automated earthquake monitoring to a new level.

## Methods

**FMNet architecture**. The neural network we design is in the category of fully convolutional network (FCN). FCN is a supervised deep learning network mainly based on convolutional layers but without fully connected layers and it has the

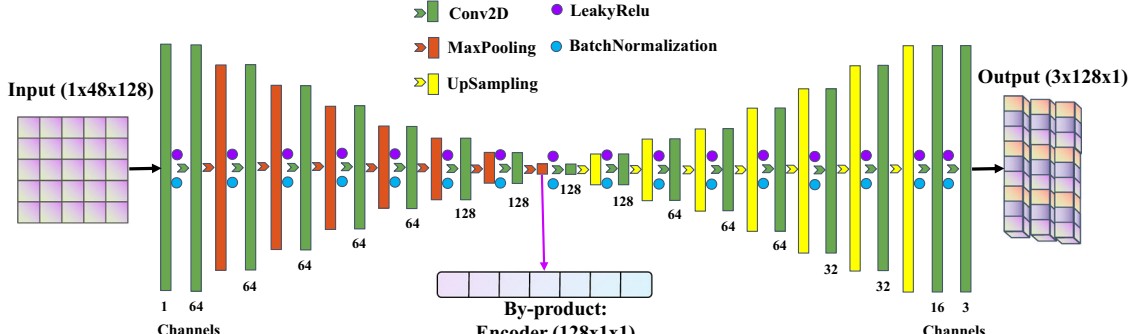

**Fig. 6 The FMNet architecture.** The designed FMNet contains 16 trainable layers as well as MaxPooling, UpSampling, LeakyRelu, and BatchNormalization layers. The input data have the size of 1 (channel of input) × 48 (3-C waveforms of 16 seismic stations) × 128 (trace length). The left part of the FMNet gradually compresses the input data from 1 × 48 × 128 to 128 × 1 × 1, and then the right part of the FMNet gradually expands the extracted features from 128 × 1 × 1 to 3 × 128 × 1 as the output of three Gaussian probability distribution representing the three angles of the focal mechanism. The intermediate layer of the encoder, with the size of 128 × 1 × 1, is also exported as a by-product to interpret the neural network.

merit of fewer model parameters and high computing efficiency[57]. Figure 6 shows the architecture of the FMNet containing a bunch of Convolutional layers, Max-Pooling layers, UpSampling layers, and also with some necessary operations such as LeakyRelu and BatchNormalization (Supplementary Software). There are 16 trainable layers. The input to the FMNet is a two-dimensional (2D) array representing the 3-C waveforms from 16 seismic stations. The output of the FMNet is three Gaussian probability distributions of the three angles of the focal mechanism. FMNet consists of two parts: a compression part that extracts the features of the input waveforms and an expansion part that transforms the extracted features to yield the output label of the focal mechanism.

In the compression part, the FMNet gradually compresses the input data from the size of 48 (3 components × 16 stations) × 128 (data length) to 1 × 1, by downsampling the input data layer by layer. At the same time, the number of filter channels gradually increases from 1 (channel of input) to 128. The data size has been changed from 1 (channel of input) × 48 (3 components × 16 stations) × 128 (data length) to 128 (channels after compression) × 1 × 1. The compression part of the FMNet can be regarded as an encoder process that compresses the data size by a factor of 48 times. The encoder is also exported as a by-product in this study for interpreting the FMNet. The encoder can take any waveform as input and output the extracted data features with the size of 128 × 1 × 1 (Supplementary Fig. 18). On the contrary, in the extension part, the FMNet gradually expands the extracted features by upsampling the features layer by layer. The upsampling operation is only carried out in the first dimension of the data, thus the size of the data in each channel is expanded from 1 × 1 to be 128 × 1. Meanwhile, the number of filter channels gradually decreases from 128 to 3. For the expansion part, the data size has been altered from 128 (channels after compression) × 1 × 1 to 3 (channels) × 128 (output length) × 1. Each output channel has a size of 128 × 1, representing a 1D Gaussian distribution. All layers use the same configuration when employing convolutional and pooling operations. Filter sizes are 3 × 3 for the compression part and 3 × 1 for the expansion part since we only expand along the first dimension of data.

**FMNet labeling and training parameters**. We design the network as a regression problem. The training label is three Gaussian probability distributions, in which the maximum probability of each distribution corresponds to one component of the source focal mechanism (i.e., strike, dip, and rake). Thus the output training label has a size of 3 × 128 × 1 (Supplementary Fig. 19). The predicted focal mechanism of the real data can be retrieved by finding the peak values of the three output Gaussian probability distributions. This formalization of training labels greatly helps the convergence when training the network, and the standard deviation of the Gaussian probability distribution affects the training convergence[16]. If the standard deviation of the Gaussian probability distribution is too small, the training process tends to be difficult to converge and if the standard deviation is too large, it may decrease the resolution of the outputs. After testing, we find that the standard deviation of 10° achieves a stable training convergence for our neural network and thus is used in this study.

Since we have designed the network as a regression problem, the mean square error (MSE) option is chosen as the training loss function[58]. For the training process, the Adam method[59] is tested to be effective in our FMNet and thus it is chosen as the optimizer, though other optimizers may also work decently. A total of 50 iterations with a batch size of 16 are implemented during the training. Besides, the learning rate is another crucial parameter that affects the level of final convergence[60]. We test different learning rates and keeping other factors the same (Supplementary Fig. 20). The performance of each learning rate is evaluated by the final training loss after the same training iterations. After testing, we find the learning rate of 0.001 achieves the smallest convergence and is therefore used for the training in this study.

## Data availability
The three-component (3-C) seismograms from 16 seismometers can be downloaded from the Southern California Seismic Network (SCSN) website.

## Code availability
The codes of FMNet and a demo on test data are submitted as a Supplementary file and they are also available from the corresponding author.

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

## Acknowledgements

We thank the financial support of the National Key R&D Program of China (Grant No. 2018YFC1504001) and the Institute of Earthquake Forecasting, China Earthquake Administration. We also appreciate discussions with Mark Zoback and Bill Ellsworth at Stanford, and Lupei Zhu at Saint Louis University. The open-source platform of Keras is used to design the deep learning neural network. GMT and Matlab softwares are used to generate the figures.

## Author contributions

W.K. designed the project, processed and analyzed the data. W.K. and J.Z. analyzed the results and wrote the manuscript. C.Y. helped analyze the results and revise the manuscript. All figures were originally created by W.K. Figures 2 and 6 received helpful suggestions from C.Y.

## Competing interests

The authors declare no competing interests.
