## [Peer Review File · Nature Communications]

REVIEWER COMMENTS

Reviewer #1 (Remarks to the Author):

Summary

This paper proposed an exciting approach using deep learning to determine the earthquake focal mechanism in near real-time. Using simulation generated synthetic waveforms as the training datasets for a fixed network in a region, the authors trained a CNN model to estimate the strike, dip and rake as a Gaussian distribution. The test results on the real Ridgecrest earthquakes looks great. Using the first part of the trained model (the compression part) as an encoder to compress the waveforms into sparse representation of the waveforms. The authors also tested the hypothesis the feature similarity in feature domain is equivalent to that in data domain. This opens the door for large database query using the compressed features. Overall, this paper is well written, and the proposed methods and tests are promising to be used in the future. The key point of this paper is to use the synthetics as the training data, in a sense, this encoded the geophysics knowledge into the deep learning model and then turn the geophysics problem into a pattern matching problem. By doing this, it can determine the focal mechanism faster without human intervention. It provides a nice proof of concept and applies to the real cases successfully. I only have some minor comments, and I would recommend a publication of this paper after minor revision. It will be a good contribution to the community. Qingkai Kong - Berkeley Seismology Lab. Comments

- There are also some disadvantages of using this approach that may need to expand a little bit in the paper, such as the network is fixed (not so flexible), what is the effect of the velocity model (if no available good velocity model exists in a region), and only usable for larger earthquakes (because the frequency band used). I hope the authors can expand some discussion in the paper to make this clear or provide some walk around. Therefore, I suggest the authors can do some quick tests to show the stability of the method if some stations are missing (make the approach a little flexible), such as change the number of stations, for example, if there are some stations have problems during normal operation, that the recordings are not reliable, how do the results change (assuming one or two stations have no data, replace the waveforms into zeros, etc.)? Generate some examples using different velocity model, and monitor the changes of the errors (also a better way to quantify the mis-match of the FM).
- A test of fewer stations and poor coverages may be interesting. The examples shown in the paper are very nice situation with big earthquakes, many station and good station coverage, to make the results more robust, what about the station coverage is not good, for example, all the stations are at one side of the event (this happens a lot in many places). Also, maybe add some discussions about out of network event, are you planning to retrain the a model with expanded training area, or this model can extrapolate, and to what extend?
- Maybe this is some future work, in Ridgecrest, there are many smaller earthquakes that have focal mechanisms till now (hundreds of M3.5+), in this paper, the authors only tested the 4 large events. But I think it is worth testing all these smaller events as well, and see the limit of the model at different frequency bands.
- When you generated the synthetics for training purposes, did you use a range of magnitude events on different grid? I can not find this information in the paper, please specify so that the readers can see what you did.
- In the paper, line 279, it is saying "the FMNet does not require the pre-knowledge of the location or depth of a real earthquake as long as it is within the monitoring area". But during training and testing, you did align these waveforms based on the theoretical P, therefore, I think this statement is not accurate. In real application, how do you align these waveforms? If you are using the theoretical P, then you do need the location information. I guess if you use the trigger onset instead of theoretical P, because when you form the matrix of the input data, these stations are in order, this automatically encode some information about the travel time for the later phases. But please make this clear.
- In preparing the training data, how did you add the realistic noise? Please make it clearer in the methods section. And in the paper, it is said adding a random 10s shift error, on all the

waveforms? or something else?

- Please report the training time on this particular training and the specification of the GPU (if used), usually this information will be interested in the community.
- You mentioned the standard deviation for the Gaussian distribution will affect the converge of the training, can you make this clearer?

Reviewer #2 (Remarks to the Author):

The authors propose a deep learning approach for earthquake focal mechanism determination. Estimating the focal mechanism of an earthquake is of interest in order to understand its physical characteristics, in particular regarding local stress redistribution and future aftershock locations.

The area of interest (around the epicenter of the Ridgecrest earthquakes) is discretized into a 3D grid. Synthetic stations are added at the same position than stations from the CI seismic network, and synthetic waveforms are generated at each of these stations for earthquakes of different focal mechanisms. An autoencoder model, trained on about 800000 examples of such synthetic data, is tasked with retrieving the focal mechanism from the recorded waveforms (strike, dip, and rake). Once trained, the model is applied to 4 real earthquakes of the Ridgecrest sequence (of magnitude greater than 5.4).

Major comments

1) Comparison with P wave first motion estimates

line 52: 'Compared to determining other source parameters of an earthquake (i.e., origin time, location, and magnitude), the estimation of the source focal mechanism usually requires much more human interactions and it is lack of full automation and efficiency.'

Regarding existing methodologies for automatized focal mechanism determination, the authors only refer to their 2014 paper based on database search. The introduction does not describe the existing literature relative to estimates of P-wave first-motion polarity by deep learning (Ross et al, 2018; Hara et al, 2019; Uchide, 2020; among others). Earthquake focal mechanisms are relatively straight-forward to derive from P-wave first motions. In particular, Ross et al. (2018) show that automated, CNN-based first motion estimations lead to much improved focal mechanisms in California compared to existing catalogs.

P-wave first motion estimates by deep learning are likely to be extremely fast. Furthermore, their major advantage is that these detections are not region dependent: the same algorithm can be applied anywhere, as it is usually based on a single-station analysis. These approaches have also been found to work well on smaller events. Therefore they appear more simple and suitable for the determination of focal mechanisms than the methodology proposed here by the authors. Given that i) the proposed approach is more complex, ii) it does not perform well on small earthquakes, and iii) it is region dependent (a 3D grid of the area of interest is used to train the model), the paper lacks a comparison with these existing automatized polarity methodologies to show what the relative strengths of their approach are (in particular in terms of timeliness).

2) Timeliness of estimates

The authors report the computation time of the focal mechanism estimates (about 200 milliseconds). This is not what matters for applications in early warning. Indeed, estimating an earthquake's focal mechanism will require that i) the waves reach the seismic stations, and ii) that the data is processed. Therefore in real scenarios, the timing to get a focal mechanism estimation after the occurrence of an earthquake will be much larger, likely of the order of several tens of seconds. This is not analyzed at all in the paper.

3) Advantage of synthetic data

The argument that a model trained on synthetic data is better in 'scenarios without enough historical source focal mechanisms for training the neural network model, especially for those regions with limited seismicity but having the potential seismic hazards', only holds if the performance of models trained on real data generalizes poorly to regions outside of the training area. This does not seem to be the case: Hara et al. (2019) shows that a model trained to estimate P-wave first motions transfers well to other regions, even without finetuning. In general, CNNs built for picking tend to generalize very well, as the task is relatively simple.

4) Lack of test set on synthetic data

The performance of the model is only shown for testing and validating data (Figures 2 and 3 of the Supplementary). There does not seem to be any testing set on synthetic data. It would be useful to report the model's performance on a real test set instead.

5) Diversity in the examples of real earthquakes

Given that the four examples analyzed have nearly identical focal mechanisms, it is difficult to assess whether this approach would work well in general. Specifically, many damaging earthquakes occur in subduction areas where the mechanisms are not strike-slip as those analyzed here, and where seismic stations can be farther away from the epicenter (as many earthquakes occur offshore). It is unclear whether the approach would work in such cases.

Minor comments

1) l. 54: 'and lacks full automation and efficiency'.

2) Figure 2: When you say that the model 'output[s] the earthquake focal mechanism directly', it would be useful to show that this output corresponds to distributions of strike, dip, and rake. This figure is not very clear.

3) l. 116: 'the heterogeneity of velocity media'.

4) l. 117: 'we generate realistic scenarios'.

5) l. 175: 'by adopting an idea developed for face recognition, where the network learns a mapping from face images to a compact Euclidean space where distances directly correspond to a measure of face similarity, we output the extracted features to analyze the reliability and robustness of our model'.

6) l. 284, l40 of the Supplementary: 'Variations in earthquake depth'.

7) l. 336: 'A total of 50 iterations with a batch size of 16'.

8) l. 43: of the Supplementary: 'The current FMNet struggles to recognize the tiny differences'

We gratefully thank the comments and suggestions from reviewer #1. Following the comments and suggestions, we have conducted more tests and made revisions to address all the comments. These comments and suggestions greatly help improve our manuscript. Also, we have prepared point-by-point responses. (- From authors)

Reviewer #1 (Remarks to the Author):

This paper proposed an exciting approach using deep learning to determine the earthquake focal mechanism in near real-time. Using simulation generated synthetic waveforms as the training datasets for a fixed network in a region, the authors trained a CNN model to estimate the strike, dip and rake as a Gaussian distribution. The test results on the real Ridgecrest earthquakes looks great. Using the first part of the trained model (the compression part) as an encoder to compress the waveforms into sparse representation of the waveforms. The authors also tested the hypothesis the feature similarity in feature domain is equivalent to that in data domain. This opens the door for large database query using the compressed features. Overall, this paper is well written, and the proposed methods and tests are promising to be used in the future. The key point of this paper is to use the synthetics as the training data, in a sense, this encoded the geophysics knowledge into the deep learning model and then turn the geophysics problem into a pattern matching problem. By doing this, it can determine the focal mechanism faster without human intervention. It provides a nice proof of concept and applies to the real cases successfully. I only have some minor comments, and I would recommend a publication of this paper after minor revision. It will be a good contribution to the community. Qingkai Kong - Berkeley Seismology Lab.

Comment 1:

There are also some disadvantages of using this approach that may need to expand a little bit in the paper, such as the network is fixed (not so flexible), what is the effect of the velocity model (if no available good velocity model exists in a region), and

only usable for larger earthquakes (because the frequency band used). I hope the authors can expand some discussion in the paper to make this clear or provide some walk around. Therefore, I suggest the authors can do some quick tests to show the stability of the method if some stations are missing (make the approach a little flexible), such as change the number of stations, for example, if there are some stations have problems during normal operation, that the recordings are not reliable, how do the results change (assuming one or two stations have no data, replace the waveforms into zeros, etc.)? Generate some examples using different velocity model, and monitor the changes of the errors (also a better way to quantify the mis-match of the FM).

Authors: Thank you very much for your helpful comments and suggestions. Following the comments and suggestions, we have added several tests to report the performance of our model in the revised manuscript:

1. In the first test, we generate a new test dataset of 1,000 unseen synthetic samples with a diversity of focal mechanisms (Supplementary Fig. S4) to test and report the model performance. This new test dataset is generated at a variety of random locations within the study area. We also add realistic noises from real recordings and picking errors into this new test data. After prediction on this new test dataset, we define a successful prediction only if the maximum values of the predicted Gaussian probability is larger than 0.7 for each test sample. With this threshold, 91.04% of the test samples can be successfully recalled. Also, we adopt the Kagan angle analysis (Kagan 1997; 2001), in which each Kagan angle quantitatively characterizes the difference in rotation angle between the true and the predicted focal mechanisms, to evaluate the estimation errors. The Kagan angle distribution results (Supplementary Fig. S5) show that a majority of the Kagan angles are within 10° as well as some Kagan angles reach 25° . We report the model performance from this test in the *Result section*:

“To evaluate the general performance and the estimation errors of our model, we have prepared another test dataset of 1,000 unseen synthetic samples

simulated with a diversity of focal mechanisms of normal, strike-slip, and reverse faulting mechanisms (Supplementary Fig. S4). Using the well-trained model, we predict the focal mechanisms on the new test dataset. We define a successful prediction only if the maximum values of the predicted Gaussian probability is larger than 0.7 for each test sample. With this threshold, 91.04% of the test samples can be successfully recalled. For these recalled events, we adopt the Kagan angle analysis^{48, 49}, in which each Kagan angle quantitatively characterizes the difference in rotation angle between the true and the predicted focal mechanisms, to evaluate the estimation errors. From the results of Kagan angle distribution (Supplementary Fig. S5), we find that a majority of the Kagan angles are within 10° as well as some Kagan angles reach 25°. This test shows that our model can stably recall most events on testing a variety of unseen data with acceptable estimation errors. Furthermore, this test also validates that our model has learned the general ability to predict a diversity of focal mechanisms.”

2. In the second test, we assume that two stations have recording problems and their waveform signals are missing (zero amplitudes as shown in Supplementary Fig. S13). Then we predict the focal mechanism using our network model. From the test results (Supplementary Fig. S14), we find our model can produce Gaussian probability distributions similar to the true distributions. This indicates that missing data at a few stations does not affect the prediction results very much. And we report this test in the *Discussion section*:

“To further verify our model on the cases with outliers, we test the scenario that some of the recording stations have data issues and waveforms are missing, but the azimuthal coverage is still good (Supplementary Fig. S13). We find that the predicted probability distributions can match well with the true distribution in terms of their shape and maximum values when partial data are missing (Supplementary Fig. S14).”

3. In the third test, we generate a new test example using a different velocity model

(Supplementary Fig. S7). We perturb the true velocity model by a maximum of 10 percent in each layer to generate the perturbed velocity model. From the prediction results (Supplementary Fig. S8), we find that the inaccurate velocity model will increase the estimation errors. We think this is because we train the neural network associated with a particular velocity model and it is sort of model-dependent. And we report this test in the *Discussion section*:

“In the Supplementary materials, we present a numerical study using a velocity model with perturbations (Supplementary Fig. S7). We perturb the true velocity model by a maximum of 10 percent in each layer to generate a different model. From the testing results (Supplementary Fig. S8), we can tell that the estimation errors for dip and rake are 12° and 30°, respectively, and their prediction probabilities are lowered as well.”

We thank you very much for these helpful comments and hope our new tests and revisions can address your comments!

Comment 2:

A test of fewer stations and poor coverage may be interesting. The examples shown in the paper are very nice situation with big earthquakes, many station and good station coverage, to make the results more robust, what about the station coverage is not good, for example, all the stations are at one side of the event (this happens a lot in many places). Also, maybe add some discussions about out of network event, are you planning to retrain the model with expanded training area, or this model can extrapolate, and to what extend?

Authors: Thank you very much for your helpful comments and suggestions. Following your suggestions, we have performed two more experiments in the revised manuscript:

1. In the first test, we halve the available stations and put them on one side of the event (Supplementary Fig. S11). From the test results (Supplementary Fig. S12), we find that the predicted probability distributions differ from the true labels in

terms of both shape and the maximum values. Two secondary local peaks in strike appear and the maximum values are lower. This test shows that a poor azimuthal coverage of stations will increase the estimation errors compared to a good azimuth coverage. This is because the azimuthal coverage, which provides the constraints for the source radiation pattern of the focal sphere, definitely affects the constraints to the focal mechanism. And we report this test in the *Discussion section*:

“We design a test with the distribution of the halved number of stations on one side of the event and the rest of stations with zero traces for data (Supplementary Fig. S11). The event is assumed to occur in an area with training data available. From the test results, we find that both strike and dip are well resolved, but the rake angle is off by nearly 30°, and the prediction probability of rake is significantly lower (about 0.5) (Supplementary Fig. S12). Therefore, it is important to evaluate the prediction probabilities.”

2. In the second test, we design an event that occurs out of the study area (Supplementary Fig. S15). From the test result (Supplementary Fig. S16), we find that the predicted Gaussian probability distributions tend to have smaller maximum values (about 0.6) than the true distributions. Although this test shows only one example, we can use the predicted maximum probability to evaluate the reliability of the predicted results. And we report this test in the *Discussion section*:

“We also test a case where an event occurs out of the study area (Supplementary Fig. S15). The test results show that the predicted probability is much smaller (about 0.6), which can help quantify the reliability of the predicted results (Supplementary Fig. S16).”

We gratefully thank you for all these test suggestions. They greatly help improve our manuscript!

Comment 3:

Maybe this is some future work, in Ridgecrest, there are many smaller earthquakes that have focal mechanisms till now (hundreds of M3.5+), in this paper, the authors only tested the 4 large events. But I think it is worth testing all these smaller events as well, and see the limit of the model at different frequency bands.

Response: Thank you very much for this insightful comment. Yes, we agree that the extension to even smaller earthquakes (M3.5+) would be very meaningful and useful and we will consider this in our future study. We have tried to process the real waveforms (alignment, bandpass filtering, and normalization) on several smaller earthquakes (M4). But we find that the waveforms of real data are mainly dominated by noise within the selected frequency band and therefore the results are not promising. Increasing the frequency band would require a finer 3D velocity model. In our future effort, we will work on smaller earthquakes (M3.5+) with a high-resolution 3D velocity model and an efficient waveform modeling tool. Following your suggestions, we add more illustration about this in the *Discussion section*:

“Since we use synthetics associated with a 1-D velocity model to create a dataset for training and testing, it limits the application to low-frequency data, which are generally available from moderate and large events.” And, “Further development efforts are needed to combine the P-wave first motions and waveform data to handle smaller events. Generating a 3-D velocity model with great details could help model the high-frequency data as well.”

We shall enroll these efforts into our next research plans and we gratefully thank you again for this very insightful comment!

Comment 4:

When you generated the synthetics for training purposes, did you use a range of magnitude events on different grid? I cannot find this information in the paper, please specify so that the readers can see what you did.

Authors: We do not consider the magnitude information, which is eliminated in the normalization step for each data sample. To specify this information, we have added

texts in the *Result section*: “*Since we normalize the waveforms of each synthetic sample based on the maximum amplitude, we choose a fixed magnitude for all events when modeling the synthetic waveforms.*”

Comment 5:

In the paper, line 279, it is saying "the FMNet does not require the pre-knowledge of the location or depth of a real earthquake as long as it is within the monitoring area". But during training and testing, you did align these waveforms based on the theoretical P, therefore, I think this statement is not accurate. In real application, how do you align these waveforms? If you are using the theoretical P, then you do need the location information. I guess if you use the trigger onset instead of theoretical P, because when you form the matrix of the input data, these stations are in order, this automatically encode some information about the travel time for the later phases. But please make this clear.

Authors: Yes, we use the picked onset of P-waves to align the waveforms. To make it clear, we have added this information in the *Result section*: “*For real data, we need take the picked onset time of each trace for carrying out the waveform alignment.*”

Comment 6:

In preparing the training data, how did you add the realistic noise? Please make it clearer in the methods section. And in the paper, it is said adding a random 10s shift error, on all the waveforms? or something else?

Authors: To clarify, we have added the following information in the *Result section*: “*The realistic noises are extracted from the real recordings at each seismic station. The random time shifts are added to each trace of the training samples to account for the picking errors.*”

Comment 7:

Please report the training time on this particular training and the specification of the GPU (if used), usually this information will be interested in the community.

Authors: Thank you very much for your suggestion. We have added this information in the *Result section*: “*The training process takes about 5 hours with 4 GPUs of NVIDIA Tesla V100 for acceleration.*”

Comment 8:

You mentioned the standard deviation for the Gaussian distribution will affect the converge of the training, can you make this clearer?

Authors: The effect of the standard deviation for Gaussian distribution is discussed in the supplementary materials of the study by Zhang et al. (2020) (previous citation numbering about this paper could be incorrect, we have corrected it). To make it clearer, we have added the following information in the *Method section*: “*If the standard deviation of the Gaussian probability distribution is too small, the training process tends to be difficult to converge and if the standard deviation is too large, it may decrease the resolution of the outputs*”

References:

1. Kagan, Y. Y. 3-D rotation of double-couple earthquake sources. *Geophysical Journal International* 106, 709-716 (1991).
2. Kagan, Y. Y. Simplified algorithms for calculation of double-couple rotation. *Geophysical Journal International* 171, 411-418 (2007).
3. Zhang, X., Zhang, J., Yuan, C., Liu, S., Chen, Z., & Li, W. Locating induced earthquakes with a network of seismic stations in Oklahoma via a deep learning method. *Scientific Reports* 10(1), 1-12 (2020).

We gratefully thank the very helpful comments and suggestions from reviewer #2. Following the comments and suggestions, we have made substantial efforts to address all the comments by adding more test results, figures, and descriptions. These comments and suggestions greatly help improve our manuscript. Also, we have prepared point-by-point responses. (- From authors).

Reviewer #2 (Remarks to the Author):

The authors propose a deep learning approach for earthquake focal mechanism determination. Estimating the focal mechanism of an earthquake is of interest in order to understand its physical characteristics, in particular regarding local stress redistribution and future aftershock locations. The area of interest (around the epicenter of the Ridgecrest earthquakes) is discretized into a 3D grid. Synthetic stations are added at the same position than stations from the CI seismic network, and synthetic waveforms are generated at each of these stations for earthquakes of different focal mechanisms. An autoencoder model, trained on about 800000 examples of such synthetic data, is tasked with retrieving the focal mechanism from the recorded waveforms (strike, dip, and rake). Once trained, the model is applied to 4 real earthquakes of the Ridgecrest sequence (of magnitude greater than 5.4).

Major comments:

Comment 1: Comparison with P wave first motion estimates

line 52: 'Compared to determining other source parameters of an earthquake (i.e., origin time, location, and magnitude), the estimation of the source focal mechanism usually requires much more human interactions and it is lack of full automation and efficiency.'

Regarding existing methodologies for automatized focal mechanism determination, the authors only refer to their 2014 paper based on database search. The introduction does not describe the existing literature relative to estimates of P-wave first-motion polarity by deep learning (Ross et al, 2018; Hara et al, 2019; Uchide, 2020; among

others). Earthquake focal mechanisms are relatively straight-forward to derive from P-wave first motions. In particular, Ross et al. (2018) show that automated, CNN-based first motion estimations lead to much improved focal mechanisms in California compared to existing catalogs.

P-wave first motion estimates by deep learning are likely to be extremely fast. Furthermore, their major advantage is that these detections are not region dependent: the same algorithm can be applied anywhere, as it is usually based on a single-station analysis. These approaches have also been found to work well on smaller events. Therefore they appear more simple and suitable for the determination of focal mechanisms than the methodology proposed here by the authors. Given that i) the proposed approach is more complex, ii) it does not perform well on small earthquakes, and iii) it is region dependent (a 3D grid of the area of interest is used to train the model), the paper lacks a comparison with these existing automatized polarity methodologies to show what the relative strengths of their approach are (in particular in terms of timeliness).

Authors: Thank you very much for your comments and suggestions. We agree that the combination of the P-wave first motion estimates via deep learning and an inversion method is an effective and fast research tool, especially for small earthquakes with less low-frequency signals. We have added more references and descriptions about the recent developments on the P-wave first motion estimates via deep learning (Ross et al., 2018; Hara et al., 2019; Uchide, 2020; Tian et al., 2020) in the *Introduction section*.

We also conducted more tests and would like to further address some of the concerns. Using the P-wave first motions to invert for the source focal mechanism is a classical approach, which includes two steps: 1) the first motion estimation; 2) and the focal mechanism inversion using the estimated first motions. The first step (first motion estimation) has been greatly improved by the recent deep learning approaches (Ross et al., 2018; Hara et al., 2019; Uchide, 2020; Tian et al., 2020). One of our published efforts was to extend the single station polarity estimation to multiple stations via deep learning (Tian et al., 2020). The second step of the approach for obtaining the focal mechanism is still based on the P-wave first motion inversion method such as the

widely accepted HASH program (Hardebeck and Shearer 2002), which was also applied by Ross et al. (2018). Aside from the deep learning approach, utilizing the P-wave first motion data takes only the up or down sign of the first-motion polarities of the recorded waveforms, therefore, the input recordings must offer sufficient azimuthal station coverage to address the severe non-uniqueness problem (Hardebeck and Shearer 2002). However, theoretically and practically, inverting three-component waveform recordings even from a single station can well resolve the source focal mechanism, this is because the ratio of SV to SH energy is dependent on source orientation (Fan and Wallace, 1991; Dreger and Helmberger, 1993; Walter, 1993; Zhao and Helmberger, 1993; Zhu and Helmberger, 1997). That explains why using waveform data to invert the focal mechanism has become a common practice in seismology (Zhao and Helmberger 1994; Zhu and Helmberger 1996; Li et al. 2011; Zhu and Ben-Zion 2013; Ross et al. 2015; Kuang et al. 2017). Accordingly, we revised *Introduction*, *Discussion*, and also added a section in *Supplementary materials* with new testing and comparison results.

We revised the *Introduction* as follows:

“Several recent studies first apply deep learning to estimate the P-wave first-motion polarities^{9, 39-41}, and then apply the first motions to carry out focal mechanism inversion using programs such as HASH⁴². One of such efforts leads to improved focal mechanisms in California compared to existing catalogs⁹. Several seismological studies also suggest that utilizing waveform data can provide better constraints for deriving the focal mechanism than using the P-wave first-motion polarities^{37,38,43}. Our objective is to develop a seamless real-time solution for obtaining the focal mechanism in an automated fashion. Directly obtaining the focal mechanism of an event from waveform data with processing effort as little as possible is more appealing.”

We revised the *Discussion* as follows:

“Different from the approach using the P-wave first motions which requires

sufficient azimuthal station coverage (Supplementary Fig. S9 and Fig. S10), three-component waveform data from a single seismic station should be able to resolve the entire source focal mechanism in theory⁵³⁻⁵⁶. However, the deep learning approach is different from inversion. When some of the stations do not offer data for input, missing data are replaced by zeroes as input while the training data still keep the full set.” And, “Further development efforts are needed to combine the P-wave first motions and waveform data to handle smaller events. Generating a 3-D velocity model with great details could help model the high-frequency data as well.”

We have also designed a new test to compare the performance between the P-wave first motion method and the FMNet method. In *Supplementary materials*, we have added the following Section 6 about this test:

6. Comparing with the P-wave first motion method

“In this study, we compare the performance of the inversion method using the P-wave first motion and our FMNet using the three-component full waveforms for resolving the source focal mechanisms. We conduct this comparison by varying the number of recording stations (Fig. S9). We use the widely accepted HASH program³ as the P-wave first motion inversion method and assume all the P-wave first motion data are correctly identified for input data. For the P-wave first motion method, we start from 6 stations and gradually increase to 30 stations. After fine-tuning a few parameters (such as minimum number of polarities, maximum azimuthal gap, number of trial inversions, the probability threshold for multiples, etc.), the HASH program performs a number of trial inversions for the same data (we set 50 inversions) and outputs a preferred solution³. We use this preferred solution as the final output of the P-wave first motion method for comparison. For FMNet, we start from 6 stations but test up to 16 stations since our model is trained with 16 stations. When the number of stations is smaller than 16, we drop several stations by setting the waveforms to zero. After we derive the focal mechanisms from both methods, we compare the results to the true focal mechanisms and calculate their Kagan angles^{1,2}, in which each Kagan angle can quantitatively characterize the difference in rotation angle between the true and

the predicted focal mechanisms, to quantify the estimation errors of each method. The comparison results show that, with the same number of stations, the P-wave first motion method (in red) consistently shows larger estimation errors (Kagan angles) compared to the proposed FMNet method (in blue) (Fig. S10). Specifically, to reduce the estimation error (Kagan angle) to about 10°, the P-wave first motion method (in red) requires 22 stations, as a contrast, the FMNet (in blue) only needs 16 stations. Therefore, compared to the P-wave first motion method, our FMNet is more straightforward and it can better constrain the source focal mechanism when fewer stations are available. It has been well reported that inverting three-component waveform data from one or two stations can well constrain the source focal mechanism⁴⁻⁷. ”

In addition to the differences in data contribution, please also note that our objective in this study is to develop a seamless real-time method for obtaining the source focal mechanism in an automated fashion. Conducting numerical inversions often requires fine-tuning parameters and quality control, which may be challenging in real-time. On the other hand, the FMNet approach seems to involve more efforts in the training data preparation and testing phase, but straight-forward when dealing with a new entry event.

Hope our revisions, new comparison test, and explanations could help clarify this concern.

Comment 2: Timeliness of estimates

The authors report the computation time of the focal mechanism estimates (about 200 milliseconds). This is not what matters for applications in early warning. Indeed, estimating an earthquake's focal mechanism will require that i) the waves reach the seismic stations, and ii) that the data is processed. Therefore in real scenarios, the timing to get a focal mechanism estimation after the occurrence of an earthquake will be much larger, likely of the order of several tens of seconds. This is not analyzed at all in the paper.

Authors: Thank you very much for your comments. Indeed the computation time that we refer to is the true computation time, which does not include the time waiting for data after the occurrence of an earthquake. We have been operating a fully AI-based earthquake monitoring system namely “EarthX” in Sichuan and Yunnan, China since 2018. A real-time report from the system is like the following (an event just occurred on September 30, 2020, local time in China):

- Origin Time: 2020-09-30 02:49:08.35
- Location: Lon=104.3 Lat=27.95
- Magnitude: ML=4.2; Mw=4
- Event Depth: Hw=10 (km)
- Source Focal Mechanism Solution:
- Fault Plane 1: Strike=330 Dip=75 Rake=0
- Fault Plane 2: Strike=240 Dip=90 Rake=165

The data waiting time depends on where events occur and the network coverage. In this particular case, it takes over a minute to collect sufficient data needed for performing eight AI analyses, which take about 11 seconds in total for conducting denoise, detection, magnitude, location, focal mechanism, depth check, outlier check, cross check, confidence analysis, and reporting. Among those, the AI computation for the focal mechanism takes about 200 ms. However, on another denser local network that

we are testing recently, the full waveform data waiting time varies from 20-40 sec.

We have added a paragraph in the *Discussion section* to clarify:

“In earthquake monitoring, it may take several tens of seconds to over a minute for receiving the full waveform data needed at a number of seismic stations depending on the source-station distance. From that point, our neural network system takes about additional 200 millisecond for analysis, which is negligible.”

Comment 3: Advantage of synthetic data

The argument that a model trained on synthetic data is better in 'scenarios without enough historical source focal mechanisms for training the neural network model, especially for those regions with limited seismicity but having the potential seismic hazards', only holds if the performance of models trained on real data generalizes poorly to regions outside of the training area. This does not seem to be the case: Hara et al. (2019) shows that a model trained to estimate P-wave first motions transfers well to other regions, even without finetuning. In general, CNNs built for picking tend to generalize very well, as the task is relatively simple.

Response: Thank you very much for your comment.

First, there have been two types of machine learning efforts in seismology: one is global and transferable to other seismic networks, the other is more focusing on a specific network. In general, those methods without using the Green's functions and recording geometry are transferrable, such as denoise, detection, picking, and first motion estimates. Those associated with Green's functions and/or recording geometry are designed for specific monitoring networks and regions. However, we must say both efforts are significant because they both help advance the state-of-the-art technologies for monitoring earthquakes.

Second, it is not likely synthetics can be used as training data for solving every deep learning problems. However, using synthetics with the Green's functions to solve the source focal mechanism has been a standard in seismology, such as the Cut and Past method (CAP, Zhao and Helmberger 1994; Zhu and Helmberger 1996), full moment tensor inversion (Zhu and Ben-Zion 2013; Ross et al. 2015), or other full waveform

matching technique (Li et al. 2011; Kuang et al. 2017). Therefore, using a deep learning approach to facilitate synthetic matching for a particular region is logical and robust at least for obtaining the focal mechanism.

In addition, generating synthetics using a velocity model is not a complicated effort. For different regions and recording networks, the approach can be repeated easily using the same programs and once the data preparation is completed, the process is straight-forward for processing any new event.

Comment 4: Lack of test set on synthetic data

The performance of the model is only shown for testing and validating data (Figures 2 and 3 of the Supplementary). There does not seem to be any testing set on synthetic data. It would be useful to report the model's performance on a real test set instead.

Authors: Thank you very much for your comments and suggestions.

Following your suggestion, we have added a new test on 1,000 unseen synthetic samples with a diversity of focal mechanisms (see Supplementary Fig. S4) to report the model's performance. This new test dataset is generated at a variety of random locations within the study area. Also, we add realistic noise from real recordings and picking errors into this new test data. We adopt the Kagan angle analysis (Kagan 1991; 2007), in which each Kagan angle quantitatively measures the difference in rotation angle between the true and the predicted focal mechanisms, to evaluate the estimation errors. The test results (Supplementary Fig. S5) show that our model has a high recall rate (>90%) and our model is very stable on testing different kinds of focal mechanisms. We have reported the model's performance from this test in the *Result section* as the following:

“To evaluate the general performance and the estimation errors of our model, we generate another test dataset of 1,000 unseen synthetic samples simulated with a diversity of focal mechanisms of normal, strike-slip, and reverse faulting mechanisms (Supplementary Fig. S4). Using the well-trained model, we predict the focal mechanisms on the new test dataset. We define a successful prediction only if the maximum values of the predicted probability distributions are larger than 0.7 for each

test sample. With this threshold, 91.04% of the test samples can be successfully recalled. For these recalled events, we adopt the Kagan angle analysis^{48, 49} to quantify the estimation errors, in which each Kagan angle quantitatively characterizes the difference in rotation angle between the true and the predicted focal mechanisms. From the results of Kagan angle distribution (Supplementary Fig. S5), we find that a majority of the Kagan angles are within 10° as well as some Kagan angles reach 25°. This test shows that our model can stably recall most events on testing a variety of unseen data with acceptable estimation errors. Furthermore, this test also validates that our model has learned the general ability to predict a diversity of focal mechanisms.”

Comment 5: Diversity in the examples of real earthquakes

Given that the four examples analyzed have nearly identical focal mechanisms, it is difficult to assess whether this approach would work well in general. Specifically, many damaging earthquakes occur in subduction areas where the mechanisms are not strike-slip as those analyzed here, and where seismic stations can be farther away from the epicenter (as many earthquakes occur offshore). It is unclear whether the approach would work in such cases.

Authors: Theoretically, our approach can handle any focal mechanisms because of using synthetics for training. To illustrate the strength of the approach on that aspect, we have added three more tests in the revised manuscript to address various scenarios:

1. In the first test, we design a new test dataset containing a diversity of focal mechanisms (normal, strike-slip, and reverse faulting mechanism) to verify the general ability of our model on predicting different kinds of focal mechanisms (this test is also for Comment 4). Adopting the Kagan angle analysis, the results (Supplementary Fig. S5) show that our model can successfully predict a diversity of focal mechanisms with reasonable estimation errors (mostly <10° and maximum of 25°). This new test validates the generalization ability of our model on predicting a diversity of focal mechanisms and also quantitatively shows the estimation errors.
2. In the second test, we assume that two stations have recording problems and their

waveform signals are missing (zero amplitudes as shown in Supplementary Fig. S13). Then we predict the focal mechanism using our network model. From the test results (Supplementary Fig. S14), we find our model can produce Gaussian probability distributions similar to the true distributions. This indicates that missing data at a few stations does not affect the prediction results very much. And we report this test in the *Discussion section*:

“To further verify our model on the cases with outliers, we test the scenario that some of the recording stations have data issues and waveforms are missing, but the azimuthal coverage is still good (Supplementary Fig. S13). We find that the predicted probability distributions can match well with the true distribution in terms of their shape and maximum values when partial data are missing (Supplementary Fig. S14).”

3. In the third test, we design an event that occurs out of the study area (Supplementary Fig. S15). From the test result (Supplementary Fig. S16), we find that the predicted Gaussian probability distributions tend to have smaller maximum values (about 0.6) than the true distributions. Although this test shows only one example, we can use the predicted maximum probability to evaluate the reliability of the predicted results. And we report this test in the *Discussion section*:

“We also test a case where an event occurs out of the study area (Supplementary Fig. S15). The test results show that the predicted probability is much smaller (about 0.6), which can help quantify the reliability of the predicted results (Supplementary Fig. S16).”

Minor comments:

Comment 6: l. 54: 'and lacks full automation and efficiency'.

Authors: Following your suggestion, we have corrected this sentence in the revised manuscript.

Comment 7: Figure 2: When you say that the model 'output[s] the earthquake focal

mechanism directly', it would be useful to show that this output corresponds to distributions of strike, dip, and rake. This figure is not very clear.

Authors: Following your suggestion, we have added more explanation in Figure 2 to show the output corresponds to strike, dip, and rake.

Comment 8: l. 116: 'the heterogeneity of velocity media'.

Authors: Thanks for the suggestion, we have corrected this sentence in the revised manuscript.

Comment 9: l. 117: 'we generate realistic scenarios'.

Authors: Thanks for the comment, we have corrected this sentence in the revised manuscript.

Comment 10: l. 175: 'by adopting an idea developed for face recognition, where the network learns a mapping from face images to a compact Euclidean space where distances directly correspond to a measure of face similarity, we output the extracted features to analyze the reliability and robustness of our model'.

Authors: Thank you very much for helping reorganize this long sentence. We have corrected this sentence in the revised manuscript and it is now much clearer.

Comment 11: l. 284, 140 of the Supplementary: 'Variations in earthquake depth'.

Authors: Following the suggestion, we have corrected this sentence in the revised manuscript.

Comment 12: l. 336: 'A total of 50 iterations with a batch size of 16'.

Authors: Thanks for the comment, we have corrected this sentence in the revised manuscript.

Comment 13: l. 43: of the Supplementary: 'The current FMNet struggles to recognize the tiny differences'

Authors: Thanks a lot for this suggestion, we have corrected this sentence in the revised manuscript.

Reference:

1. Ross, Z. E., Meier, M. A. & Hauksson, E. P wave arrival picking and first - motion polarity determination with deep learning. *Journal of Geophysical Research: Solid Earth* 123(6), 5120-5129 (2018).
2. Uchide, T. Focal mechanisms of small earthquakes beneath the Japanese islands based on first-motion polarities picked using deep learning. *Geophysical Journal International* 223(3), 1658-1671 (2020).
3. Hara, S., Fukahata, Y. & Iio, Y. P-wave first-motion polarity determination of waveform data in western Japan using deep learning. *Earth, Planets and Space* 71(1), 127 (2019).
4. Tian, X., Zhang, W., Zhang, X., Zhang, J., Zhang, Q., Wang, X. & Guo, Q. Comparison of Single - Trace and Multiple - Trace Polarity Determination for Surface Microseismic Data Using Deep Learning. *Seismological Research Letters* 91(3), 794-1803 (2020).
5. Hardebeck, J. L. & Shearer, P. M. A new method for determining first-motion focal mechanisms. *Bulletin of the Seismological Society of America* 92(6), 2264-2276 (2002).
6. Fan, G. & Wallace, T. The determination of source parameters for small earthquakes from a single, very broadband seismic station. *Geophysical Research Letters* 18(8), 1385-1388 (1991).
7. Dreger, D. S. & Helmberger, D. V. Determination of source parameters at regional distances with three - component sparse network data. *Journal of Geophysical Research: Solid Earth* 98(B5), 8107-8125 (1993).
8. Walter, W. R. Source parameters of the June 29, 1992 Little Skull Mountain earthquake from complete regional waveforms at a single station. *Geophysical Research Letters* 20(5), 403-406 (1993).
9. Zhao, L. & Helmberger, D. V. Source retrieval from broadband regional

- seismograms: Hindu Kush region. *Physics of the earth and planetary interiors* 78(1-2), 69-95 (1993).
10. Zhu, L., Helmberger, D.V., Saikia, C.K. & Woods, B. B. Regional waveform calibration in the Pamir - Hindu Kush region. *Journal of Geophysical Research: Solid Earth* 102(B10), 22799-22813 (1997).
 11. Zhao, L. S. & Helmberger, D. V. Source estimation from broadband regional seismograms. *Bulletin of the Seismological Society of America* 84, 91–104 (1994).
 12. Zhu, L. & Helmberger, D. V. Advancement in source estimation techniques using broadband regional seismograms. *Bulletin of the Seismological Society of America* 86, 1634–1641 (1996).
 13. Li, J., Zhang, H., Sadi Kuleli, H. & Nafi Toksoz, M. Focal mechanism determination using high-frequency waveform matching and its application to small magnitude induced earthquakes. *Geophysical Journal International* 184(3), 1261-1274 (2011).
 14. Zhu, L. & Ben-Zion, Y. Parametrization of general seismic potency and moment tensors for source inversion of seismic waveform data. *Geophysical Journal International* 194(2), 839-843 (2013).
 15. Ross, Z.E., Ben-Zion, Y. & Zhu, L. Isotropic source terms of San Jacinto fault zone earthquakes based on waveform inversions with a generalized CAP method. *Geophysical Journal International* 200(2), 1269-1280 (2015).
 16. Kuang, W., Zoback, M. & Zhang, J. Estimating geomechanical parameters from microseismic plane focal mechanisms recorded during multistage hydraulic fracturing. *Geophysics* 82(1), KS1-KS11 (2017).
 17. Kagan, Y. Y. 3-D rotation of double-couple earthquake sources. *Geophysical Journal International* 106, 709-716 (1991).
 18. Kagan, Y. Y. Simplified algorithms for calculation of double-couple rotation. *Geophysical Journal International* 171, 411-418 (2007).

REVIEWER COMMENTS

Reviewer #1 (Remarks to the Author):

Thanks to the authors addressed my comments and added more tests to improve the paper.

Overall, the authors answered all my comments with more tests and discussions in the paper, I only have a few follow-up comments based on this and hope the authors can answer and test.

* Regarding the answers to my comment 1, since the authors have already done the tests with dropping stations, are these dropped stations randomly selected? If yes, please specify in the paper.

* Also, the authors showed that when perturbing the velocity model, or the out of network events test, the performance of the model does degrade, it is better to clearly specify these limitations in the discussion instead of just list the results, unless this can be addressed.

* Based on the answer to my comment 4, the authors used fixed magnitude for generating the training samples, which may introduce problems. Though the authors normalized the waveform, which reduced the effect of amplitude, there are more factors that change when the magnitude varies, such as duration of the waveform, SNR, for example. For a waveform based method instead of only using the first motion polarity, I think this will have an effect, it is better to study this well. Especially the real test results only show on a few big events, it is hard to evaluate these aspects. My concern is that the trained model only tuned to estimate the results well on a very limited range of events, but in reality, you do have various cases of magnitude events that make the simulation more complicated.

* Regarding the application of the model, I do agree with the other reviewer that the model currently still limited, i.e. depend on the region, network, and only works on large earthquakes. Hope the authors can continue to make improvement of the model in the future.

Reviewer #2 (Remarks to the Author):

Many thanks to the authors for their comments, and detailed additions that helped a lot to improve and clarify the manuscript. The modifications are very precise and well explained, with several additional paragraphs and figures in Supplementary to illustrate and quantify the tests conducted. However I have some concerns regarding the results in testing. The previous version of the manuscript did not include any measure of model performance, and I am puzzled by those now added in the test analysis.

In what follows, comments from the authors are in brackets, and responses are not.

Comment 1: Comparison with P wave first motion estimates.

« We revised the Introduction as follows [...] »

Thank you very much for the added literature references, which are useful to put the proposed methodology in perspective compared to other existing methods. The paragraph added to the introduction conducts a detailed analysis of the literature relative to automated P wave first motion estimates, and the description of waveform-based versus first motion estimates of focal mechanism is also very helpful in the context of this paper.

« We revised the Discussion as follows [...] »

Many thanks for the added paragraphs.

«We have also designed a new test to compare the performance between the P-wave first motion method and the FMNet method.»

Many thanks for the additional tests which are of great interest in order to emphasize the strength of the proposed approach. Indeed the algorithm presented here seems to outperform first arrivals methods (with lower estimation errors), and to perform better when the number of stations is low. Something I'm curious about is the relative computation time of both approaches. Even in the

absence of finetuning, the proposed methodology is likely to be much faster - do you have an overall idea of the speed difference? Is the finetuning a time-consuming exercise? Overall I find these tests very convincing to highlight the advantages of the algorithm as a real-time estimator of earthquake focal mechanism.

Comment 2: Timeliness of estimates.

« We have added a paragraph in the Discussion section to clarify: [...] »

Thank you for the additional paragraph in the Discussion, which I believe is important to mention in the context of real-time applications.

Comment 3: Advantage of synthetic data.

Many thanks for the explanation. Indeed, the use of synthetic data can be straight-forward and adequate to train a model, and does appear so for large earthquakes in the present study.

However there are clear limitations here for small earthquakes (page 15: 'it is challenging to model the high-frequency theoretical waveforms with a simple 1D velocity model'). Adding a small caveat on the use of synthetic data for training the model would be useful.

Comment 4: Lack of test set on synthetic data.

« Following your suggestion, we have added a new test on 1,000 unseen synthetic samples with a diversity of focal mechanisms [...] »

Many thanks to the authors for generating a real test set. However I do not understand how the performance metrics were chosen. First, why discard all examples where the maximum probability is less than 0.7? Second, why report a recall score, that is an evaluation metric used for classification problems? This a regression setting and recall is not adapted here.

No evaluation of the model was provided in the original manuscript. Since the evaluation of the model presented in the new Supplementary paragraphs was strange, I took a look at the code and re-ran it on the test set provided. Computing the Kagan angles on the test set led to quite different results than those provided in Figure S5. In particular, while Figure S5 is cut at 25 degrees, there appears to be heavy tails in the distribution (Figure attached). More than 10% of the estimates have an error larger than 20 degrees, which seems high for a model trained and tested on synthetic data. The fraction of errors above 50 or 60 degrees is also far from negligible. Therefore while the model is probably faster than existing methods, I'm not sure that one could argue that it outperforms them in terms of precision. Looking at a few individual examples, it is likely that a symmetry issue is impacting the estimations of the neural network.

Comment 5: Diversity in the examples of real earthquakes.

« To illustrate the strength of the approach on that aspect, we have added three more tests in the revised manuscript to address various scenarios [...] »

Many thanks for the added tests and figures.

We gratefully thank the comments and suggestions from reviewer #1. Following the comments and suggestions, we have made revisions to address all the comments. Also, we have prepared point-by-point responses. (- From authors)

Reviewer #1 (Remarks to the Author):

Thanks to the authors addressed my comments and added more tests to improve the paper.

Overall, the authors answered all my comments with more tests and discussions in the paper, I only have a few follow-up comments based on this and hope the authors can answer and test.

Comment 1:

Regarding the answers to my comment 1, since the authors have already done the tests with dropping stations, are these dropped stations randomly selected? If yes, please specify in the paper.

Authors: Yes, these dropped stations are randomly selected. We have added this information in the *Supplementary materials Section 8*. “*In such a case, we randomly select two recording stations and replace the waveforms with zeroes*”.

Comment 2:

Also, the authors showed that when perturbing the velocity model, or the out of network events test, the performance of the model does degrade, it is better to clearly specify these limitations in the discussion instead of just list the results, unless this can be addressed.

Authors: Thank you very much for your comment. We have revised the *Discussion* to specify these limitations. “*From these test results, we find that inaccurate velocity model, poor azimuthal coverage, or events out of the network might degrade the prediction performance with low probability. Therefore, using the predicted probability to quantify the reliability of the predicted result is essential.*”

Comment 3:

Based on the answer to my comment 4, the authors used fixed magnitude for generating the training samples, which may introduce problems. Though the authors normalized the waveform, which reduced the effect of amplitude, there are more factors that change when the magnitude varies, such as duration of the waveform, SNR, for example. For a waveform based method instead of only using the first motion polarity, I think this will have an effect, it is better to study this well. Especially the real test results only show on a few big events, it is hard to evaluate these aspects. My concern is that the trained model only tuned to estimate the results well on a very limited range of events, but in reality, you do have various cases of magnitude events that make the simulation more complicated.

Authors: Thank you very much for your comment. We agree that earthquakes with different magnitudes will have different source durations of waveforms and might present different SNRs. When preparing the training data, we have considered these two factors to mitigate their effect.

First, although different source durations affect the shape of the full-band waveforms, after filtering the waveforms into a specific frequency range (0.05 Hz - 0.1 Hz), we find the differences between the waveforms simulated using different source durations are minor (see the figure below). Therefore, the bandpass filtering step should help reduce the effect of different source durations concerning different magnitudes.

Second, to account for different SNRs, in our implementation, we have added real noise from real recordings by randomly scaling the amplitudes of noise. By doing so, we are able to account for the effect of different SNRs. Now we have clarified this information in *the Result section* of our new manuscript. “*When adding the realistic noise, we randomly scale the amplitudes of noise to account for different Signal-to-Noise-Ratios (SNR).*”

Comment 4:

Regarding the application of the model, I do agree with the other reviewer that the model currently still limited, i.e. depend on the region, network, and only works on large earthquakes. Hope the authors can continue to make improvement of the model in the future.

Authors: Thank you very much for this comment. Following the comment, we specify the current limitations of the model in the *Discussion section* of the new manuscript. “*The current FMNet is designed for monitoring local or regional events within the coverage of a seismic network. Similar to the state-of-the-art methodology for resolving source focal mechanisms by applying moment tensor inversion, the FMNet is limited to moderate and large earthquakes that can be numerically modeled. Developing the capability to simulate waveforms of small earthquakes in high frequency warrants further study.*”

We gratefully thank the very helpful comments and suggestions from reviewer #2. Following the comments and suggestions, we have made substantial efforts to address all the comments. These comments greatly help improve our manuscript. Also, we have prepared point-by-point responses. (- From authors).

Reviewer #2 (Remarks to the Author):

Many thanks to the authors for their comments, and detailed additions that helped a lot to improve and clarify the manuscript. The modifications are very precise and well explained, with several additional paragraphs and figures in Supplementary to illustrate and quantify the tests conducted. However I have some concerns regarding the results in testing. The previous version of the manuscript did not include any measure of model performance, and I am puzzled by those now added in the test analysis.

In what follows, comments from the authors are in brackets, and responses are not.

Comment 1: Comparison with P wave first motion estimates.

«We revised the Introduction as follows [...] »

Thank you very much for the added literature references, which are useful to put the proposed methodology in perspective compared to other existing methods. The paragraph added to the introduction conducts a detailed analysis of the literature relative to automated P wave first motion estimates, and the description of waveform-based versus first motion estimates of focal mechanism is also very helpful in the context of this paper.

«We revised the Discussion as follows [...] »

Many thanks for the added paragraphs.

«We have also designed a new test to compare the performance between the P-wave first motion method and the FMNet method. »

Many thanks for the additional tests which are of great interest in order to emphasize the strength of the proposed approach. Indeed the algorithm presented here seems to outperform first arrivals methods (with lower estimation errors), and to perform better when the number of stations is low. Something I'm curious about is the relative

computation time of both approaches. Even in the absence of finetuning, the proposed methodology is likely to be much faster - do you have an overall idea of the speed difference? Is the finetuning a time-consuming exercise? Overall I find these tests very convincing to highlight the advantages of the algorithm as a real-time estimator of earthquake focal mechanism.

Authors: We are glad that our previous revisions addressed your comments. The finetuning of our FMNet is involved in the training process and it might take hours to days. However, after the network is well-trained, it does not require further finetuning when it is applied to a real earthquake. The first-motion-based inversion method tends to involve finetuning of the inversion parameters for each new event but the FMNet is straightforward in prediction and it takes less than one second to retrieve the focal solution.

Comment 2: Timeliness of estimates.

«We have added a paragraph in the Discussion section to clarify: [...] »

Thank you for the additional paragraph in the Discussion, which I believe is important to mention in the context of real-time applications.

Authors: Thank you.

Comment 3: Advantage of synthetic data.

Many thanks for the explanation. Indeed, the use of synthetic data can be straightforward and adequate to train a model, and does appear so for large earthquakes in the present study. However there are clear limitations here for small earthquakes (page 15: ‘it is challenging to model the high-frequency theoretical waveforms with a simple 1D velocity model’). Adding a small caveat on the use of synthetic data for training the model would be useful.

Authors: We agree with you on the limitations here for small earthquakes and we have specified these limitations in the manuscript. This is also the limitation in all of the existing methods when adopting waveform matching with a simplified 1D velocity model for source focal mechanism inversions in the current seismology. We will need

to further develop the modeling capability for small earthquakes. Further modeling the high-frequency theoretical waveforms will require an accurate 3D velocity model and an efficient modeling tool with tremendous computational efforts (such as Wang and Zhan, 2020). To ease this concern, we specify the current limitations of the model in the *Discussion section* of the new manuscript. *“The current FMNet is designed for monitoring local or regional events within the coverage of a seismic network. Similar to the state-of-the-art methodology for resolving source focal mechanisms by applying moment tensor inversion, the FMNet is limited to moderate and large earthquakes that can be numerically modeled. Developing the capability to simulate waveforms of small earthquakes in high frequency warrants further study.”*

Comment 4: Lack of test set on synthetic data.

« Following your suggestion, we have added a new test on 1,000 unseen synthetic samples with a diversity of focal mechanisms [...] »

Many thanks to the authors for generating a real test set. However I do not understand how the performance metrics were chosen. First, why discard all examples where the maximum probability is less than 0.7? Second, why report a recall score, that is an evaluation metric used for classification problems? This a regression setting and recall is not adapted here.

Authors: Thank you for your comment.

First, using the maximum value of the predicted probability to evaluate the confidence level of each prediction is essential. Because the maximum value of the predicted probability can help quantify the uncertainty of the results. Using a threshold to evaluate the prediction reliability has been commonly adopted by deep learning studies (such as Zhu and Beroza 2018; Zhang et al. 2020; Mousavi et al. 2020). For example, Zhu and Beroza (2018) set the threshold of probability to 0.5 for both P and S picks. Zhang et al. (2020) eliminate false results using a preset threshold of the probability value. Mousavi et al. (2020) output the deep learning results if at least one phase (P or S) with a probability above a specified threshold value exists.

Second, we agree that the use of the recall score for this regression problem may

not be appropriate. Therefore, following your suggestion, we have omitted the use of the recall score in the revised manuscript.

No evaluation of the model was provided in the original manuscript. Since the evaluation of the model presented in the new Supplementary paragraphs was strange, I took a look at the code and re-ran it on the test set provided. Computing the Kagan angles on the test set led to quite different results than those provided in Figure S5. In particular, while Figure S5 is cut at 25 degrees, there appears to be heavy tails in the distribution (Figure attached). More than 10% of the estimates have an error larger than 20 degrees, which seems high for a model trained and tested on synthetic data. The fraction of errors above 50 or 60 degrees is also far from negligible. Therefore while the model is probably faster than existing methods, I'm not sure that one could argue that it outperforms them in terms of precision. Looking at a few individual examples, it is likely that a symmetry issue is impacting the estimations of the neural network.

Figure provided by Reviewer #2

Authors: Thank you very much for testing our codes. Following your comments, we have carefully investigated this test and we would like to clarify in the following:

1. This test consists of two steps: FMNet prediction and Kagan angle calculation. We have repeated this test and found that the FMNet predictions (the first step) from our results and your results are the same. However, when calculating the Kagan angles (the second step), we used the formula Equation (28) in Kagan's paper (2007) (which only works for Kagan angle $< 90^\circ$), but we should use the generalized formula Equation (29) in that paper (which works for Kagan angle $< 120^\circ$). Now we have corrected this mistake and the recalculated Kagan angle distribution as shown below (Figure R1) is consistent with your results but with possible minor differences due to different bin sizes used for plotting the histogram. Therefore, except for the Kagan angle calculation (the second step, now it has been corrected), the FMNet prediction code is working consistently between our results and your results.

Figure R1. Test performance of the previous FMNet model

2. From this new plot, we do notice that there are about 10% of the estimates show errors larger than 20° as you indicated. We found that was due to the random time shifts added to data too large: a maximum of 10 seconds for simulating picking

errors in data. Indeed this is unnecessarily too large. Further review of the literature on picking suggests random picking error up to 2 seconds is more than sufficient (Zhu & Beroza 2018; Ross et al. 2018; Mousavi et al. 2020).

For example, results from Zhu and Beroza (2018) show picking errors mostly within 0.1 sec. Ross et al. (2018) report that the arrival time picks from deep learning are within 0.028 sec of the analyst picks; Mousavi et al. (2020) show the standard deviation of picking errors is 0.03 sec with the attentive deep-learning model (Earthquake Transformer). Therefore, now we have reduced the random time shifts from a maximum of 10 sec to a maximum of 2 sec for re-preparing both the training and the test data. This will change both the training and test datasets in terms of random time shifts. Using the new training dataset, we have retrained our FMNet model while keeping all other parameters (such as network architecture, learning rate, iterations, and so on) to be the same.

After we finish the retraining, we test the performance of the improved FMNet model on the new test dataset (with smaller random time shifts, provided in the newly submitted codes). Then, we repeat the Kagan angle analysis for the improved FMNet model. The figure below (Figure R2) shows the test performance of our improved FMNet model on the new test data. With this improved model, we find that about 97.8% of the estimates show errors smaller than 20° , and only a small fraction of about 2.2% of the estimates have an error larger than 20° . Investigating the remaining 2.2%, we find it is probably caused by the equivalent property of the two nodal planes of the focal mechanism (the true and the auxiliary nodal planes are equivalent). We also want to point out that, further finetuning and improving the model would be possible. For example, increasing the size of training data can help improve the model performance. Adding more constraints such as first-motion polarities should also help mitigate the tail. The improved FMNet model and the new test data are provided in the new submission. In the new submission, to ensure you can get the same results as ours, we have also included our predicted results and our Kagan angle calculation code for your reference. Please kindly follow the detailed steps specified in the “README” file when you

run the codes.

Figure R2. Test performance of the improved FMNet model

3. Since the improved FMNet model shows improved test performance, to ensure the consistency of our manuscript, we have also taken this opportunity to re-examine this improved FMNet model on both the real data application and other tests. Using the improved FMNet model, we redo both the real data application and other tests. From the new results, we find that overall the new results and conclusions remain consistent with the previous results and conclusions, although some new results show minor improvements. This is reasonable and logical because reducing the random time shifts of training data should not affect other tests (for example, the test for missing data, the test for velocity errors, the test for recording azimuthal coverage, and the test for an event out of the network) much. To ensure the consistency of our manuscript, we have replaced relevant figures using the new results from the improved FMNet model, although the differences are very minor. To be specific, we list the minor changes of several figures here: 1), the training and validation loss curve of Fig. S2 in the supplementary materials; 2), the real data

application results of Fig.3 in the main manuscript; 3), the test performance of the improved FMNet model on the synthetic test dataset of Fig. S5 in the supplementary materials; 4), the test for velocity errors of Fig. S8 in the supplementary materials; 5), the test for poor azimuthal coverage of Fig. S12 in the supplementary materials; 6), the test for missing data of Fig. S14 in the supplementary materials; 7), the test for an event out of the network of Fig. S16 in the supplementary materials.

These minor changes for the improved FMNet model do not affect our main results and conclusions.

4. Finally, we also have gone through the full manuscript and made some text revisions to ensure the text consistency of our study. Mainly there are two text revisions: a), text descriptions about the test performance of the improved FMNet model; b), text descriptions about other tests. We list the text revisions as follows:

- a) We specify the random time shifts that are added to account for picking errors are now within 2 seconds in the data preparation section. And we have modified the text descriptions about the test performance of the improved FMNet model as follows:

“To evaluate the general performance and the estimation errors of our model, we generate another test dataset of about 1,000 unseen synthetic samples simulated with a diversity of focal mechanisms of normal, strike-slip, and reverse faulting mechanisms (Supplementary Fig. S4). Using the well-trained model, we predict the focal mechanisms on this test dataset. For these predicted focal mechanisms, we adopt the Kagan angle analysis^{48, 49} to quantify the estimation errors, in which each Kagan angle quantitatively characterizes the difference in rotation angle between the true and the predicted focal mechanisms. From the results of Kagan angle distribution (Supplementary Fig. S5), we find that 97.8% of the Kagan angles are within 20° and only a small fraction of about 2.2% of the estimates have an error larger than 20°.

Investigating the remaining 2.2%, we find it is likely caused by the equivalent property of the two nodal planes of the focal mechanism (the true and the auxiliary nodal planes are equivalent). Including more constraints such as first-motion polarities should possibly mitigate this issue and further improve the model. Nevertheless, this test shows that our model can stably predict most events (97.8%) on testing a variety of unseen data with acceptable estimation errors. Furthermore, this test also validates that our model has learned the general ability to predict a diversity of focal mechanisms.”

b) Other minor text modifications about other tests. Text descriptions for other tests mostly remain consistent and unchanged. Only very minor modifications are made from the new results and they do not affect our discussions and conclusions:

- i. “From the testing results (Supplementary Fig. S8), we can tell that the estimation errors for dip and rake are about ~~12°~~8° and ~~30°~~20°, respectively.”
- ii. “From the test results, we find that both the strike and dip angle are well resolved, but the rake angles is off by nearly ~~30°~~20°, and the prediction probability of rake is significantly lower (about 0.5) (Supplementary Fig. S12).”

To briefly summarize, following your comments, we have corrected the mistake in the Kagan angle calculation and we have improved the FMNet model. Using the improved FMNet model, we also re-examine the real data application and other tests to ensure the consistency of our manuscript. At last, we want to gratefully thank all your comments. These comments indeed greatly help improve the strength and completeness of our manuscript. Also, we hope our responses and revisions can address your comments and concerns.

Comment 5: Diversity in the examples of real earthquakes.

« To illustrate the strength of the approach on that aspect, we have added three more

tests in the revised manuscript to address various scenarios [...] »

Many thanks for the added tests and figures.

Authors: We are glad that you are happy with our previous revisions.

Reference:

1. Wang, X. & Zhan, Z. Seismotectonics and Fault Geometries of the 2019 Ridgecrest Sequence: Insight From Aftershock Moment Tensor Catalog Using 3 - D Green's Functions. *Journal of Geophysical Research: Solid Earth* 125(5), e2020JB019577 (2020).
2. Zhu W. & Beroza G. C. PhaseNet: a deep-neural-network-based seismic arrival-time picking method. *Geophysical Journal International*, 216(1), pp.261-273 (2018).
3. Zhang, X., Zhang, J., Yuan, C., Liu, S., Chen, Z., & Li, W. Locating induced earthquakes with a network of seismic stations in Oklahoma via a deep learning method. *Scientific reports* 10(1), 1-12 (2020).
4. Mousavi, S.M., Ellsworth, W.L., Zhu, W., Chuang, L.Y. & Beroza, G.C. Earthquake transformer—an attentive deep-learning model for simultaneous earthquake detection and phase picking. *Nature Communications* 11(1), pp.1-12 (2020).
5. Kagan, Y. Y. Simplified algorithms for calculation of double-couple rotation. *Geophysical Journal International* 171, 411-418 (2007).
6. Ross, Z. E., Meier, M. A. & Hauksson, E. P wave arrival picking and first - motion polarity determination with deep learning. *Journal of Geophysical Research: Solid Earth* 123(6), 5120-5129 (2018).

REVIEWERS' COMMENTS

Reviewer #1 (Remarks to the Author):

The authors addressed all my comments, and I think the paper is in good shape. I only have 1 minor comment

Minor comment

In the discussion session, the following sentence not clear, please consider modifying it to make it clear.

"Different from the approach using the P-wave first motions which requires sufficient azimuthal station coverage (Supplementary Fig. S9 and Fig. S10), inverting three-component waveform data from a single seismic station should be able to resolve the entire source focal mechanism in theory. However, the deep learning approach is different from inversion....."

Reviewer #2 (Remarks to the Author):

The results presented in the new version of the manuscript are based on extensive changes made to the model. I'm glad that my suggestions helped to improve the model's performance and robustness. The new results appear much more reliable than before. In what follows, comments from the authors are in brackets ({}) and my answers are not.

{However, when calculating the Kagan angles [...] we should use the generalized formula Equation (29) in that paper (which works for Kagan angle $< 120^\circ$). Now we have corrected this mistake and the recalculated Kagan angle distribution as shown below (Figure R1) is consistent with your results but with possible minor differences due to different bin sizes used for plotting the histogram.}

I was very puzzled by this discrepancy – I now understand the cause of the difference between the previous version of the paper and my tests of the code.

{From this new plot, we do notice that there are about 10% of the estimates show errors larger than 20° as you indicated. We found that was due to the random time shifts added to data too large [...] Therefore, now we have reduced the random time shifts from a maximum of 10 sec to a maximum of 2 sec for re-preparing both the training and the test data. [...] After we finish the retraining, we test the performance of the improved FMNet model on the new test dataset [...]. With this improved model, we find that about 97.8% of the estimates show errors smaller than 20° , and only a small fraction of about 2.2% of the estimates have an error larger than 20° .}

Indeed a 2 seconds picking error seems to be reasonable. The results from the new model appear much more reliable and robust, in particular regarding the much smaller tails in the error distribution.

{Investigating the remaining 2.2%, we find it is probably caused by the equivalent property of the two nodal planes of the focal mechanism (the true and the auxiliary nodal planes are equivalent). We also want to point out that, further finetuning and improving the model would be possible.}

Just a small suggestion for future work, in case it might be useful: setting the Kagan angle as loss function of the neural network (instead of MSE as it is now) should be helpful to further reduce these types of errors.

{However, after the network is well-trained, it does not require further finetuning when it is applied to a real earthquake. [...]}

By finetuning, I wasn't referring to the finetuning of the model but to the process of adjusting the HASH standard algorithm that you described in the previous changes to the manuscript: «After fine-tuning a few parameters (such as minimum number of polarities, maximum azimuthal gap, number of trial inversions, the probability threshold for multiples, etc.), the HASH program performs a number of trial inversions for the same data (we set 50 inversions) and outputs a preferred solution». During the last modifications, you added a comparison of the errors between your approach and the existing HASH methods, in Figure S10. If it is rapid to run and not too much work, I was wondering whether a similar comparison would be possible in terms of speed. This would highlight a lot the strength of the of the FMNet versus classic algorithms in an early warning setting.

Overall, the results with the new model appear much more convincing than in the previous version, and answer my concerns.

We thank the comments from reviewer #1. Following the comments, we have made revision to address the comments. (- From authors)

Reviewer #1 (Remarks to the Author):

The authors addressed all my comments, and I think the paper is in good shape. I only have 1 minor comment

Minor comment

In the discussion session, the following sentence not clear, please consider modifying it to make it clear.

"Different from the approach using the P-wave first motions which requires sufficient azimuthal station coverage (Supplementary Fig. S9 and Fig. S10), inverting three-component waveform data from a single seismic station should be able to resolve the entire source focal mechanism in theory. However, the deep learning approach is different from inversion....."

Authors: Thank you. We have modified this sentence as follows: *“Approach using the P-wave first motions requires sufficient azimuthal station coverage to constrain the focal sphere, however, inverting three-component waveform data from a single seismic station should be able to resolve the entire source focal mechanism in theory⁵³⁻⁵⁶. The FMNet utilizes the three-component full waveform information and it shows smaller estimation errors compared to the approach based on the P-wave first motions (Supplementary Fig. S9 and Fig. S10).”*

We thank the comments from reviewer #2. Accordingly, we have addressed all the comments as follows. (- From authors).

Reviewer #2 (Remarks to the Author):

The results presented in the new version of the manuscript are based on extensive changes made to the model. I'm glad that my suggestions helped to improve the model's performance and robustness. The new results appear much more reliable than

before. In what follows, comments from the authors are in brackets ({}) and my answers are not.

{However, when calculating the Kagan angles [...] we should use the generalized formula Equation (29) in that paper (which works for Kagan angle $< 120^\circ$). Now we have corrected this mistake and the recalculated Kagan angle distribution as shown below (Figure R1) is consistent with your results but with possible minor differences due to different bin sizes used for plotting the histogram. }

I was very puzzled by this discrepancy – I now understand the cause of the difference between the previous version of the paper and my tests of the code.

Authors: We are glad that our previous revisions addressed your comments. Thank you.

{From this new plot, we do notice that there are about 10% of the estimates show errors larger than 20° as you indicated. We found that was due to the random time shifts added to data too large [...] Therefore, now we have reduced the random time shifts from a maximum of 10 sec to a maximum of 2 sec for re-preparing both the training and the test data. [...] After we finish the retraining, we test the performance of the improved FMNet model on the new test dataset [...]. With this improved model, we find that about 97.8% of the estimates show errors smaller than 20° , and only a small fraction of about 2.2% of the estimates have an error larger than 20° . }

Indeed a 2 seconds picking error seems to be reasonable. The results from the new model appear much more reliable and robust, in particular regarding the much smaller tails in the error distribution.

Authors: Thank you.

{Investigating the remaining 2.2%, we find it is probably caused by the equivalent property of the two nodal planes of the focal mechanism (the true and the auxiliary

nodal planes are equivalent). We also want to point out that, further finetuning and improving the model would be possible.}

Just a small suggestion for future work, in case it might be useful: setting the Kagan angle as loss function of the neural network (instead of MSE as it is now) should be helpful to further reduce these types of errors.

Authors: Thank you very much for this suggestion. We will consider this suggestion in our future work.

{However, after the network is well-trained, it does not require further finetuning when it is applied to a real earthquake. [...]}

By finetuning, I wasn't referring to the finetuning of the model but to the process of adjusting the HASH standard algorithm that you described in the previous changes to the manuscript: «After fine-tuning a few parameters (such as minimum number of polarities, maximum azimuthal gap, number of trial inversions, the probability threshold for multiples, etc.), the HASH program performs a number of trial inversions for the same data (we set 50 inversions) and outputs a preferred solution». During the last modifications, you added a comparison of the errors between your approach and the existing HASH methods, in Figure S10. If it is rapid to run and not too much work, I was wondering whether a similar comparison would be possible in terms of speed. This would highlight a lot the strength of the of the FMNet versus classic algorithms in an early warning setting.

Authors: Thank you very much for your comment. As we addressed in the previous revision, the proposed FMNet shows smaller estimation errors than the classic P-wave first motion method, especially when only a few stations are available. To further address your comment, we conducted another test to compare the speed. From the test result, the HASH program takes about 840 ms to output a preferred solution, which is also fast compared to ours (~200ms). Therefore, both the FMNet and the P-wave first motion method are fast and they are comparable in terms of speed. But the FMNet is

more direct (one step) than the Polarity-based approach (two steps) for the final focal mechanism solution. More importantly, when only a few close stations are available, the FMNet can estimate the focal mechanism not only fast but also more accurately than the classic P-wave first motion method.

Overall, the results with the new model appear much more convincing than in the previous version, and answer my concerns.

Authors: Thank you again for all your comments and suggestions.